# Interface Junctions in QCD$_4$

Pranay Gorantla[1a] and Ho Tat Lam[1b]

[1] Physics Department, Princeton University, Princeton, NJ, USA

**Abstract**

We study 3+1 dimensional $SU(N)$ Quantum Chromodynamics (QCD) with $N_f$ degenerate quarks that have a spatially varying complex mass. It leads to a network of interfaces connected by interface junctions. We use anomaly inflow to constrain these defects. Based on the chiral Lagrangian and the conjectures on the interfaces, charaterized by a spatially varying $\theta$-parameter, we propose a low-energy description of such networks of interfaces. Interestingly, we observe that the operators in the effective field theories on the junctions can carry baryon charges, and their spin and isospin representations coincide with baryons. We also study defects, characterized by spatially varying coupling constants, in 2+1 dimensional Chern-Simons-matter theories and in a 3+1 dimensional real scalar theory.

Tuesday 14$^{\text{th}}$ July, 2020

---

[a]e-mail: `gorantla@princeton.edu`
[b]e-mail: `htlam@princeton.edu`

# 1 Introduction and Summary

A quantum field theory can form various defects by making its coupling constants space-dependent. These defects can be co-dimension one interfaces, co-dimension two strings and so on. They can also intersect and form junctions of higher co-dimensions. Sometimes there

are localized degrees of freedom on the defects with intricate dynamics. These localized degrees of freedom are often protected by anomaly inflow [1], generalized anomalies involving coupling constants [2,3] or higher berry phase [4–6].

A typical example is interfaces in 3+1 dimensional $SU(N)$ Yang-Mills theory with a position-dependent $\theta$-angle that interpolates from $\theta = 0$ to $\theta = 2\pi$.[1] The interface supports an $SU(N)_{-1}$ Chern-Simons theory on its worldvolume [7–9]. The Chern-Simons theory has a $\mathbb{Z}_N$ one-form symmetry with an 't Hooft anomaly that cancels the anomaly inflow from the bulk [8,10,11]. The anomaly inflow can also be phrased in terms of a generalized anomaly involving the $\theta$-angle [3]. The $\mathbb{Z}_N$ one-form symmetry is spontaneously broken in the Chern-Simons theory, which signals deconfinement of probe quarks on the interface.

Similar interfaces with varying $\theta$ have also been studied in 3+1 dimensional $SU(N)$ QCD [3,9,12].[2] The Lagrangian of the theory is[3]

$$\mathcal{L} = \frac{1}{4g^2}\text{Tr}\left(f^{\mu\nu}f_{\mu\nu}\right) + \frac{\theta}{8\pi^2}\epsilon^{\mu\nu\rho\sigma}\text{Tr}(f_{\mu\nu}f_{\rho\sigma}) + \sum_{I=1}^{N_f}\overline{\psi}_I(i\not{D}+m)\psi_I \ , \qquad (1.1)$$

where $f$ is the field strength of the dynamical gauge field $c$. Depending on the fermion mass $m$, the number of flavors $N_f$ and the number of colors $N$, the interface can either support a topological quantum field theory, a gapless sigma model or a trivially gapped theory. The theory on the interface is closely related to the dualities of Chern-Simons-matter theories [14].

In this note, we will explore other defects in $SU(N)$ QCD by making the complex fermion mass $m$ vary on a plane. We will use two coordinates on the plane interchangeably, the radial coordinate $(r, \alpha)$ and the Cartesian coordinate $(x, y) = (r\cos\alpha, r\sin\alpha)$. Consider a winding mass profile $m \propto re^{if(\alpha,r)}$ where $f(\alpha, r)$ is a smooth function that satisfies $f(\alpha + 2\pi, r) = f(\alpha, r) + 2\pi$. At large radius, the fermions are heavy so they can be integrated out. This reduces the theory to a pure $SU(N)$ Yang-Mills theory with a winding $\theta$-angle, $\theta = N_f f(\alpha)$. It leads to $N_f$ interfaces centered at the trajectories where $f(\alpha, r) = (\pi + 2\pi\mathbb{Z})/N_f$. Each of these interfaces supports an $SU(N)_{-1}$ Chern-Simons theory. The fermion path integral also generates a classical winding counterterm of the background gauge fields and the metric. The winding of such counterterms is robust under the addition of other smooth space-dependent counterterms.

The question we would like to answer is what happens in the interior of the space. Let us first consider the anomaly inflow from large radius. There are two contributions.

---

[1]Our discussion is restricted to smooth interfaces whose dynamics is uniquely determined by the microscopic theory. It is to be contrasted with discontinuous interfaces where $\theta$ jumps abruptly. Discontinuous interfaces have an ambiguity of adding more degrees of freedom localized on the interfaces.

[2]See [13] for a study of QCD domain walls from a holographic perspective.

[3]We will work with Lagrangians in the Lorentzian signature $(-, +, +, +)$.

One of them is the gravitational anomaly inflow from the $SU(N)_{-1}$ Chern-Simons theories on the interfaces. The other one comes from the winding counterterm. The nontrivial anomaly inflow implies that there are gapless degrees of freedom localized in the interior. We will present a coherent picture of the interior based on the chiral Lagrangian of QCD and some conjectured low-energy descriptions of the $\theta$-varying interfaces. The proposal is consistent with the anomaly inflow. For the validity of the chiral Lagrangian, our discussions throughout the paper will be restricted to $N_{\mathrm{CFT}} \geq N_f \geq 1$ where $N_{\mathrm{CFT}}$ is the lower bound of the conformal window. Despite the restriction, many discussions in this note can be applied to $N_f \geq N_{\mathrm{CFT}}$.

Let us summarize our proposal. We will first consider a specific profile $m = \varepsilon r e^{i\alpha} \Lambda^2$. Here $\Lambda$ is the QCD scale and $\varepsilon$ is a dimensionless constant. Below we will study two cases, $\varepsilon \ll 1$ and $\varepsilon \gtrsim 1$.

For $N_f = 1$, as illustrated in figure 2, the only interface at large radius continues along the radial direction and terminates around the point where the quadratic potential of the $\eta'$ particle vanishes. The interface theory undergoes a transition from the $SU(N)_{-1}$ Chern-Simons theory to a trivially gapped theory at certain radius $R_1 \sim 1/(\varepsilon\Lambda)$. The transition leads to a 1+1 dimensional compact chiral boson localized around the transition interface. Interestingly, the excitations of the chiral boson can carry baryon charges, and the spin of corresponding operator $\mathcal{O}_N = e^{iN\phi}$ coincides with the spin of the baryon $\epsilon^{a_1 \cdots a_N} \psi_{a_1} \cdots \psi_{a_2}$ in the ultraviolet theory.[4]

Around the end point of the interface, the quadratic potential of the $\eta'$ particle vanishes. One might suspect that the $\eta'$ particle gets localized around the end point and reduces to some gapless 1+1 dimensional excitataions. We will show that this intuition is not correct. On the contrary, the localized excitations of the $\eta'$ particle have a non-zero effective mass in 1+1 dimensions so there are no gapless degrees of freedom localized at the end point.

For $N_{\mathrm{CFT}} > N_f > 1$, the discussions are divided into two cases $\varepsilon \ll 1$ and $\varepsilon \gtrsim 1$.

- When $\varepsilon \ll 1$, the $N_f$ interfaces continue along the radial direction and meet at the origin where they form an interface junction of size $R_2 \sim 1/(\varepsilon^{1/3}\Lambda)$. At certain radius $R_1 \sim 1/(\varepsilon\Lambda)$, the theories on the interfaces undergo a transition from an $SU(N)_{-1}$ Chern-Simons theory to a $\mathbb{CP}^{N_f-1}$ sigma model with a Wess-Zumino term. On the junction, the fields obey an orthogonality condition (5.12) that reduces the target space from $N_f$ copies of $\mathbb{CP}^{N_f-1}$ manifold to the flag manifold

$$\frac{U(N_f)}{\prod_{a=1}^{N_f} U(1)} . \tag{1.2}$$

---

[4] This observation has been recently employed in the quantum Hall droplet proposal for the $N_f = 1$ baryons [15]. An important element of the proposal is a meta-stable sheet of the $\eta'$ particle. In contrast to the proposal, the setup we considered with a spatially varying fermion mass, although not translation invariant, is stable.

We emphasize that the flag sigma model on the junction is not an isolated 1+1 dimensional theory. It should be viewed as the boundary theory of the 2+1 dimensional theories on the interfaces. The flag sigma model can be parametrized by a $U(N_f)$ matrix with a block-diagonal $\prod_{a=1}^{N_f} U(1)$ gauge symmetry acting from the right. It is supplemented by a Wess-Zumino term $S_{\text{WZW}}$ of the $U(N_f)$ matrix field defined in (5.22). $S_{\text{WZW}}$ is not invariant under the $\prod_{a=1}^{N_f} U(1)$ gauge symmetry but its non-invariance is canceled by the gauge variation of the Wess-Zumino terms of the $\mathbb{CP}^{N_f-1}$ sigma models on the interfaces. If the flag sigma model were an isolated theory, $S_{\text{WZW}}$ would not be allowed since it is not gauge invariance. The flag sigma model on the junction and the Wess-Zumino term $S_{\text{WZW}}$ can be derived from the chiral Lagrangian. The proposal is summarized in figure 3.

- When $\varepsilon \gtrsim 1$, $R_2 \gtrsim R_1$ so the $SU(N)_{-1}$ Chern-Simons theories on the interfaces are in direct contact with the interface junction. The theory on the junction becomes a gauged $U(N_f)$ Wess-Zumino-Witten model with a restricted block-diagonal $\prod_{a=1}^{N_f} U(1)$ gauge symmetry acting from the right. The gauge parameters are restricted such that they depend only on one light coordinate. Such chirally gauged Wess-Zumino-Witten model was studied in [16]. The theory has a chiral algebra that consists of a $\mathfrak{u}(N_f)_N$ left-moving chiral algebra and a $\mathfrak{u}(N_f)_N / \prod_{a=1}^{N_f} \mathfrak{u}(1)_N$ right-moving coset chiral algebra (we do not pay attention to the global form of the chiral algebra). The chiral algebra has the correct 't Hooft anomaly that cancels the anomaly inflow from large radius. Interestingly, the junction theory has operators that carry baryon charges. For instance, it has an operator with spin $N/2$ which transforms under the $Sym^N(\square)$ representation of the $U(N_f)$ global symmetry. The spin of the operator coincides with the spin of the baryons in the same isospin representation.

For a general mass profile $m = \varepsilon r e^{if(\alpha,r)}\Lambda^2$, the interfaces still form a junction at the origin but some of them can merge into one interface before joining with other interfaces at the origin (see figure 6). This leads to a network of interfaces connected by interface junctions. Each of these junctions supports a 1+1 dimensional chiral algebra. The total central charge and the total 't Hooft anomaly of these chiral algebras are constrained by the anomaly inflow.

The low-energy descriptions above for interfaces and interface junctions are similar to the descriptions for domain walls and domain wall junctions in $\mathcal{N} = 1$ supersymmetric gauge theory [17–20]. The theory on the domain wall is conjectured to be a supersymmetric Chern-Simons theory [17] and the theory on the domain wall junction is conjectured to have a supersymmetric coset chiral algebra [18].[5] We emphasize that domain walls and interfaces are two different objects. Domain walls are dynamical excitations that separate two different

---

[5]See [21–24] for related discussions on domain wall junctions in supersymmetric theories.

vacua while interfaces are defects created by the variation of coupling constants. Similarly, domain wall junctions should be distinguished from interface junctions.

The rest of the paper is organized as follows. In section 2, we review the chiral Lagrangian and the dynamics on the interfaces with a varying $\theta$-parameter. In section 3, we initiate the study of $SU(N)$ QCD with a space-dependent fermion mass $m \propto re^{i\alpha}$ by analyzing the large radius behavior of the theory. In section 4 and 5, we propose a coherent picture of the interior for $N_f = 1$ and $N_f > 1$ respectively. In section 6, we discuss more general mass profiles $m \propto re^{if(\alpha,r)}$. Appendix A studies interfaces in Chern-Simons-matter theories including $U(1)_k$ Chern-Simons theory coupled to $N_f$ scalars with a spatially varying mass squared and $SU(N)_{-k+N_f/2}$ Chern-Simons theory coupled to $N_f$ fermions with a spatially varying mass when $N_f > k > 0$. Appendix B studies defects in a 3+1 dimensional real scalar theory defined by making the coefficients of the quadratic and the linear term in the potential position-dependent.

# 2 Background

## 2.1 Phase Diagram

We begin by reviewing the QCD phase diagram presented in [9]. The phase diagram is consistent with the large $N$ analysis [25–29] and the constraint from 't Hooft anomalies [3,8].

The theory depends on the complex fermion mass $m$ and the $\theta$-angle only through the combination $M^{N_f}$ where $M = me^{i\theta/N_f}$. The theory is trivially gapped at generic $M$. It has a first order phase transition line along the negative real axis of $M^{N_f}$ coming from infinity. The first-order phase transition is associated to the spontaneous symmetry breaking of the CP symmetry. When $N_f = 1$, the line terminates at $M = M_0 < 0$ with a massless $\eta'$ particle. When $N_f > 1$, the line terminates at $M = 0$ with an $SU(N_f)$ non-linear sigma model for $N_{\text{CFT}} > N_f > 1$, an interacting CFT for $\frac{11}{2}N > N_f > N_{\text{CFT}}$ or a free gauge theory for $N_f > \frac{11}{2}N$. The phase diagram is summarized in Fig 1.

Our discussion below will be restricted to $N_{\text{CFT}} > N_f \geq 1$. We will briefly comment on the cases with $N_f > N_{\text{CFT}}$.

### 2.1.1 $\eta'$ Effective Field Theory

For $N_f = 1$, the low energy dynamics of the $\eta'$ particle near $M = M_0$ can be described by an effective Lagrangian

$$\mathcal{L}_{\eta'} = f_{\eta'}^2 \left( \frac{1}{2}(\partial \eta')^2 + \kappa \eta' + \frac{1}{2}\mu^2 \eta'^2 + \frac{1}{4}\lambda \eta'^4 \right) , \tag{2.1}$$

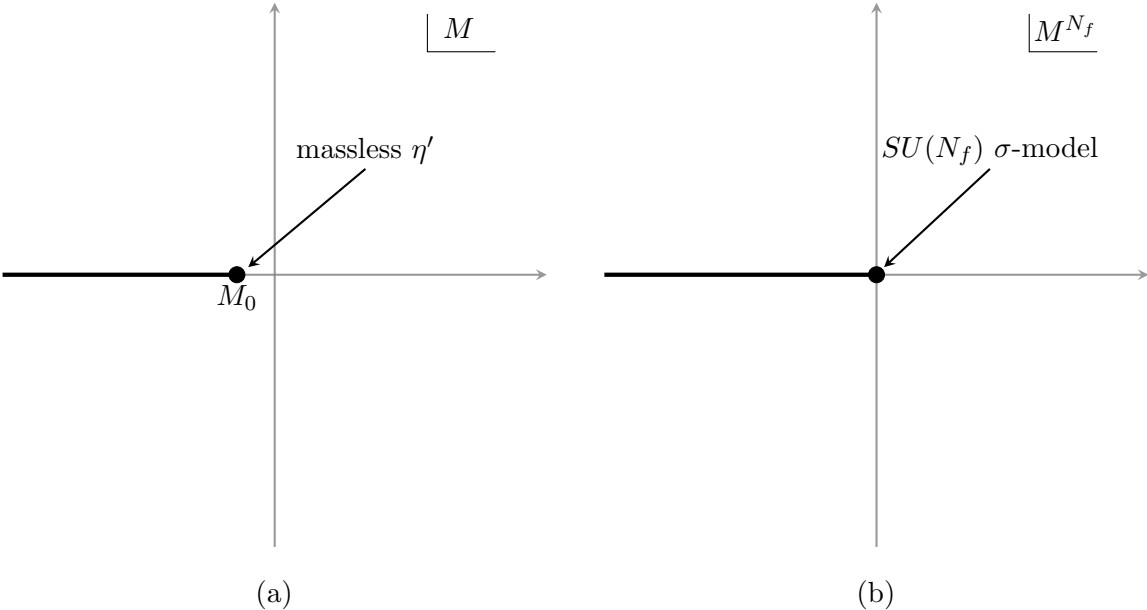

Figure 1: (a) The phase diagram of QCD with single quark. The first order line terminates at $M = M_0$ with a massless $\eta'$ particle. (b) The phase diagram of QCD with $N_{\text{CFT}} > N_f > 1$ quarks. The first order line terminates at $M = 0$ with an $SU(N_f)$ non-linear sigma model.

where $\kappa \propto \text{Im}(M)$ and $\mu^2 \propto (\text{Re}(M) - M_0)$. The $\eta'$ particle is a pseuodoscalar so when it condenses for $M < M_0$, it breaks the time-reversal symmetry.

The $\eta'$ particle has a particularly nice interpretation in the large $N$ limit. In the strictly infinite $N$ limit, the $U(1)$ axial symmetry is restored at $M = 0$ and the $\eta'$ particle can be identified with the phase of the chiral condensate

$$\langle \overline{\psi}\psi \rangle = f_{\eta'}^2 \Lambda e^{i\eta'} \ , \tag{2.2}$$

and hence, as the Nambu-Goldstone boson of the restored $U(1)$ axial symmetry. Here we define $\Lambda$ to be the QCD scale. $\Lambda$ does not scale with $N$ in the large $N$ limit but $f_{\eta'} \sim \sqrt{N}\Lambda$. Because of the Adler-Bell-Jackiw anomaly, $1/N$ correction explicitly breaks the $U(1)$ axial symmetry and correspondingly generates a small mass for the $\eta'$ particle. For $|M| \ll \Lambda$, the $\eta'$ particle can be described by an effective Lagrangian[6]

$$\mathcal{L}_{\eta'} = \frac{1}{2} f_{\eta'}^2 \left( \partial \eta' \right)^2 - f_{\eta'}^2 \Lambda (\text{Re}(M) \cos \eta' - \text{Im}(M) \sin \eta') + \frac{1}{2}\chi \Lambda^4 \text{min}_k (\eta' + 2\pi k)^2 \ . \tag{2.3}$$

The effective Lagrangian only includes the leading $1/N$ corrections. Here $\chi$ is the topological susceptibility defined by the two-point function of the instanton density in the pure gauge

---

[6]The $\eta'$ potential has a cusp singularity at $\theta = \pi$, which signals a rearrangement of the heavy fields. This means that the effective $\eta'$ field theory breaks down when $\eta'$ crosses $\pm\pi$.

theory. The $\eta'$ particle becomes massless at $M = M_0$ where $M_0 = -\chi\Lambda^3/f_{\eta'}^2$. Around $M = M_0$, the effective Lagrangian (2.3) reduces to (2.1).

### 2.1.2 $SU(N_f)$ Chiral Lagrangian

For $N_{\text{CFT}} > N_f > 1$, the chiral symmetry is spontaneously broken at $M = 0$ which leads to an $SU(N_f)$ non-linear sigma model. We will ignore the dynamics of the $\eta'$ field. The $SU(N_f)$ field is denoted by $U$. When $N_f \geq 3$, the sigma model has a Wess-Zumino term, which is defined using a five-dimensional extension of the field configuration:

$$\Gamma_{\text{WZ}} = \frac{i}{240\pi^2} \int_{N_5} \text{Tr}\left[(U^\dagger dU)^5\right] , \tag{2.4}$$

where $N_5$ is a five-manifold whose boundary is the original spacetime $M_4$. The coefficient of the Wess-Zumino term has to be an integer such that its weight $\exp(i\Gamma_{\text{WZ}})$ is independent of the extensions. In order to match with the perturabative 't Hooft anomalies of QCD, the coefficient needs to be $N$ [30]. When $N_f = 2$, the sigma model has no Wess-Zumino term, instead, it has a four-dimensional $\mathbb{Z}_2$-valued discrete $\theta$ term associated to $\pi_4(SU(2)) = \mathbb{Z}_2$ [31]. We will use the same notation $\Gamma_{\text{WZ}}$ to denote the discrete $\theta$ term. When $N_f = 2$, the ultraviolet theory has no perturbative anomaly for the $SU(2)_L \times SU(2)_R$ symmetry but it can have a non-pertubative $\mathbb{Z}_2$-valued $SU(2)$ anomaly associated to $\pi_4(SU(2)) = \mathbb{Z}_2$ [32]. To match with the non-perturbative anomaly, the coefficient of the $\theta$ term has to be $N$ mod 2.

The chiral Lagrangian is still valid for $|M| \ll \Lambda$ but it includes a potential generated by the fermion mass term. The full action is given by

$$S_\pi = \frac{1}{2}f_\pi^2 \int_{M_4} \left(\text{Tr}(\partial_\mu U^\dagger \partial^\mu U) - \Lambda\text{Tr}(MU + M^*U^\dagger)\right) + N\Gamma_{\text{WZ}} , \tag{2.5}$$

where $M = me^{i\theta/N_f}$ is defined before.

The non-linear sigma model has skyrmions associated to $\pi_3(SU(N_f)) = \mathbb{Z}$. These skyrmions are identified with the baryons in the ultraviolet theory [31]. The skyrmion current is given by

$$J_\mu = \frac{1}{24\pi^2}\epsilon_{\mu\nu\rho\sigma}\text{Tr}\left[(U^\dagger\partial^\nu U)(U^\dagger\partial^\rho U)(U^\dagger\partial^\sigma U)\right] . \tag{2.6}$$

The 't Hooft anomalies involving the baryon current in the ultraviolet theory can be reproduced by the chiral Lagrangian with the skrymion current [30].

## 2.2 Interfaces

We can create an interface in QCD by making the $\theta$-angle vary from $\theta = 0$ to $\theta = 2\pi k$ along one coordinate. Such $\theta$-varying interfaces have been discussed extensively in [3,9,11,12,19]. We will focus on the interfaces with $k = 1$.

For large fermion mass $m \gg \Lambda$, the bulk theory reduces to an $SU(N)$ Yang-Mills theory with a varying $\theta$, which leads to an $SU(N)_{-1}$ Chern-Simons theory on the interface. For small fermion mass $m \ll \Lambda$, the bulk theory reduces to the effective field theories discussed in section 2.1. For $N_f = 1$, the effective field theory of the $\eta'$ particle leads to a trivially gapped interface theory.

For $N_{\text{CFT}} > N_f > 1$, the effective field theory is an $SU(N_f)$ non-linear sigma model with a position-dependent potential

$$-\frac{1}{2}mf_\pi^2\Lambda \left(e^{i\theta(x)/N_f}\text{Tr}(U) + c.c\right) . \tag{2.7}$$

On one side, $\theta = 0$, the potential is minimized at $U = \mathbb{1}$. On the other side, $\theta = 2\pi$, the potential is minimized at $U = e^{-2\pi i/N_f}\mathbb{1}$. Without lost of generality, we can interpolate between them using a diagonal matrix

$$V = \begin{pmatrix} e^{i\varphi_1} & & & \\ & e^{i\varphi_2} & & \\ & & \ddots & \\ & & & e^{i\varphi_{N_f}} \end{pmatrix}, \quad \sum \varphi_a = 0 \bmod 2\pi . \tag{2.8}$$

The vacuum has $\varphi_1 = (1 - N_f)\varphi$ and $\varphi_2 = \cdots = \varphi_{N_f} = \varphi$ where $\varphi(x)$ is a function that interpolates from 0 to $-2\pi/N_f$. The other vacua can be generated by the $SU(N_f)$ global symmetry $U = gVg^\dagger$. The vacua constitute a

$$\mathbb{CP}^{N_f-1} = \frac{U(N_f)}{U(1) \times U(N_f - 1)} \tag{2.9}$$

manifold which then becomes the target space of the three-dimensional non-linear sigma model on the interface.

The sigma model has a three-dimensional Wess-Zimino term which descends from the Wess-Zumino term $\Gamma_{\text{WZ}}$ of the bulk chiral Lagrangian [9]. The $\mathbb{CP}^{N_f-1}$ sigma model can be parametrized by $N_f$ complex scalars $\Phi_I$ with a constraint $\sum \Phi_I^\dagger \Phi_I = 1$ and a $U(1)$ gauge symmetry $\Phi_I \to \Phi_I e^{-i\lambda}$. Alternatively, it can be parametrized by a $U(N_f)$ matrix $g_{Ia}$ with a $U(1) \times U(N_f - 1)$ block-diagonal gauge symmetry acting from the right. The two descriptions are related by $\Phi_I = g_{I1}$. The three-dimensional field on the interface and the

four-dimensional bulk field $U(x, \vec{z})$ are related by

$$U(x, \vec{z}) = g(\vec{z})V(x)g(\vec{z})^\dagger \;, \tag{2.10}$$

where $\vec{z}$ denotes the transverse coordinates on the interface. To compute the Wess-Zumino term $\Gamma_{\mathrm{WZ}}$, we can construct a five-dimensional extension of $U$ by extending only $g(\vec{z})$ to a four-manifold $N_4$. The boundary of $N_4$ is the worldvolume of the interface $M_3$. This choice of extension is only for computational convenience. The Wess-Zumino term does depend on the choice of the extensions. We notice that

$$g^\dagger U^\dagger dU g = V^\dagger g^\dagger dg V + V^\dagger dV - g^\dagger dg \;. \tag{2.11}$$

Since $V$ depends only on the $x$ coordinate, we need exactly one factor of $V^\dagger dV$ appearing in the Wess-Zumino term. The Wess-Zumino term $N\Gamma_{\mathrm{WZ}}$ then simplifies to

$$\frac{iN}{48\pi^2} \int_{N_4 \times \mathbb{R}} \mathrm{Tr}\left[V^\dagger dV (V^\dagger g^\dagger dg V - g^\dagger dg)^4\right] = \frac{C_{abcd}}{12\pi^2} \int_{N_4} (g^\dagger dg)_{ab}(g^\dagger dg)_{bc}(g^\dagger dg)_{cd}(g^\dagger dg)_{da} \;. \tag{2.12}$$

The coefficient $C_{abcd}$ is

$$\begin{aligned}
C_{abcd} &= N \int \sin\left(\frac{\varphi_{ab}}{2}\right) \sin\left(\frac{\varphi_{bc}}{2}\right) \sin\left(\frac{\varphi_{cd}}{2}\right) \sin\left(\frac{\varphi_{da}}{2}\right) d\left(\varphi_{ba} + \varphi_{dc}\right) \\
&= \frac{3\pi}{2} N \big((1 - \delta_{a1})\delta_{b1}(1 - \delta_{c1})\delta_{d1} - \delta_{a1}(1 - \delta_{b1})\delta_{c1}(1 - \delta_{d1})\big) \;,
\end{aligned} \tag{2.13}$$

where $\varphi_{ab} = \varphi_a - \varphi_b$. In the end, we obtain the three-dimensional Wess-Zumino term for the $\mathbb{CP}^{N_f - 1}$ sigma model

$$-\frac{N}{4\pi} \int_{N_4} d(g^\dagger dg)_{11} d(g^\dagger dg)_{11} = \frac{N}{4\pi} \int_{M_3 = \partial N_4} b db \;, \tag{2.14}$$

where we define a composite gauge field $b = i \sum \Phi_I^\dagger d\Phi_I = i(g^\dagger dg)_{11}$. When $N_f = 2$, the bulk theory has only a discrete $\mathbb{Z}_2$-valued $\theta$ term, which reduces to the discrete $\mathbb{Z}_2$-valued $\theta$ term of the $\mathbb{CP}^1$ non-linear sigma model associated to $\pi_3(\mathbb{CP}^1) = \mathbb{Z}$ [33, 34]. The $\theta$-term can also be presented as a Chern-Simons term of the composite gauge field $b$

$$\pi N \left(\frac{1}{4\pi^2} \int_{M_3} b db\right) \;. \tag{2.15}$$

The term in the parenthesis is an integer when $M_3$ is a closed manifolds.

Using the relation (2.10), we can also reduce the bulk skyrmion current (2.6) to the

interface:

$$\int J_\mu dx = \frac{1}{8\pi^2}\epsilon_{\mu\nu\rho}(g^\dagger\partial^\nu g)_{ab}(g^\dagger\partial^\rho g)_{ba}\int \sin^2\left(\frac{\varphi_{ab}}{2}\right)d\varphi_{ba} = \frac{1}{4\pi}\epsilon_{\mu\nu\rho}\partial^\nu b^\rho \ , \qquad (2.16)$$

where the indices are restricted to the ones for the transverse coordinates. The reduced current coincides with the skyrmion current of the $\mathbb{CP}^{N_f-1}$ sigma model associated to $\pi_2(\mathbb{CP}^{N_f-1}) = \mathbb{Z}$. This means that skyrmions in the $\mathbb{CP}^{N_f-1}$ sigma model can be interpreted as baryons localized on the interfaces.

In summary, the $k=1$ interface theory has two phases. For large fermion mass $m \gg \Lambda$, the theory is an $SU(N)_{-1}$ Chern-Simons theory. For small fermion mass $m \ll \Lambda$, the theory is trivially gapped when $N_f = 1$, a $\mathbb{CP}^{N_f-1}$ non-linear sigma model with a Wess-Zumino term when $N_{\text{CFT}} > N_f > 1$. For $N_f > N_{\text{CFT}}$, it is natural to conjecture that the interface theory remains $SU(N)_{-1}$ Chern-Simons theory for all mass [9].

For $N_{\text{CFT}} > N_f \geq 1$, the interface must go through a phase transition when the bulk fermion mass increases. Whether this phase transition is first-ordered or second-order has not been determined. A possible three-dimensional theory that captures this phase transition is $U(1)_N$ Chern-Simons theory coupled to $N_f$ scalars [9]. The theory is consistent with the anomaly inflow from the bulk [3, 9]. It has a conjectured fermionic dual, an $SU(N)_{-1+N_f/2}$ Chern-Simons theory coupled to $N_f$ fermions [14]. In the following sections, we will assume the validity of these effective interface theories.

Let us briefly summarize the dynamics on $k > 1$ interfaces. We will assume $k < N_f/2$. The theories on the interfaces depend on how fast $\theta$ varies. For large mass $m \gg \Lambda$, there is only one interface that supports an $SU(N)_{-k}$ when $|\nabla\theta| \gg \Lambda$ and $k$ separated interfaces with each of them supporting an $SU(N)_{-1}$ when $|\nabla\theta| \ll \Lambda$. For small mass $m \ll \Lambda$, the interface theory is trivially gapped for $N_f = 1$ and a non-linear sigma model for $N_{\text{CFT}} > N_f > 1$. When $|\nabla\theta| \gg \sqrt{m\Lambda}$, there is only one interface and the target space of the sigma model is the Grassmannian manifold

$$\mathcal{M}(k, N_f - k) = \frac{U(N_f)}{U(k) \times U(N_f - k)} \ . \qquad (2.17)$$

When $|\nabla\theta| \ll \sqrt{m\Lambda}$, there are $k$ interfaces and each of them supports a $\mathbb{CP}^{N_f-1}$ sigma model. As before, these non-linear sigma models have nontrivial Wess-Zumino terms (see appendix A.2 for a detailed discussion on the Wess-Zumino terms).

# 3   Large Radius Behavior and Anomaly Inflow

We now promote the complex fermion mass $m$ to be space-dependent while fixing the $\theta$-angle to be zero. The fermion mass has a winding profile $m = \varepsilon r e^{i\alpha}\Lambda^2$.[7] We will first analyze the theory at large radius and then determine the dynamics in the interior in the subsequent sections.

At large radius where $|m| \gg \Lambda$, we can integrate out the fermions. In this process, we will keep track of the classical counterterms of the background gauge fields and the metric.

The global symmetry of the theory includes a $U(N_f)/\mathbb{Z}_N$ symmetry. For simplicity, we will not couple the theory to the most general background. Instead, we turn on only an $SU(N_f) \times U(1)$ background. The background consists of a $U(1)$ gauge field $A$ with a field strength $F_A$ and an $SU(N_f)$ gauge field $B$ with a field strength $F_B$. The two gauge fields can be combined into a $U(N_f)$ gauge field $B + A\mathbb{1}$. $A$ can be viewed as the background gauge field of the baryon symmetry. It is normalized such that the baryon has charge $N$.

Before integrating out the fermions, we can remove the phase of the fermion mass $m$ by a chiral rotation. This generates a winding $\theta$-angle for the dynamical and background gauge fields as well as for the metric:[8]

$$\mathcal{L} \supset N_f \alpha \frac{\text{Tr}(f \wedge f)}{8\pi^2} + NN_f \alpha \frac{F_A \wedge F_A}{8\pi^2} + N\alpha \frac{\text{Tr}(F_B \wedge F_B)}{8\pi^2} + 2NN_f \alpha \frac{\text{Tr}(R \wedge R)}{384\pi^2} \ . \quad (3.1)$$

We emphasize that this procedure is possible only for non-zero $m$ and therefore only away from $r = 0$. After integrating out the fermions, the large radius theory becomes a pure $SU(N)$ Yang-Mills theory with $\theta = N_f \alpha$ supplemented by a winding counterterm for the background.

It is well-known that quantum field theories are subject to an ambiguity due to the addition of smooth space-dependent counterterms. Since the plane is contractible, the coefficients of smooth counterterms can never have nontrivial winding numbers around infinity. This means that the winding number of the counterterm at infinity is robust but the precise form of the counterterm can be deformed as long as the winding number is preserved. The winding counterterm provides an anomaly inflow to the interior. The anomaly inflow is characterized by an anomaly polynomial

$$- 2\pi NN_f \frac{F_A \wedge F_A}{8\pi^2} - 2\pi N \frac{\text{Tr}(F_B \wedge F_B)}{8\pi^2} - 4\pi NN_f \frac{\text{Tr}(R \wedge R)}{384\pi^2} \ , \quad (3.2)$$

which depends only on the winding number of the counterterm.

---

[7]Even though the theory depends only on $M^{N_f}$, we can not consider the fermion mass profile $m \propto re^{i\alpha/N_f}$ since it has branch cut.

[8]The integral $\frac{1}{384\pi^2}\int \text{Tr}(R \wedge R)$ is an integer on closed spin manifolds and an integer multiple of $\frac{1}{48}$ on closed non-spin manifolds. It defines the gravitational Chern-Simons term $d\text{CS}_{\text{grav}} = \frac{1}{384\pi^2}\text{Tr}(R \wedge R)$.

The winding $\theta$-angle for the dynamical gauge field, $\theta = N_f \alpha$, leads to $N_f$ interfaces centered at $\alpha = (\pi + 2\pi\mathbb{Z})/N_f$. Notice that the condition $|\nabla\theta| \ll \Lambda$ always holds at sufficiently large radius. Each of these interfaces support an $SU(N)_{-1}$ Chern-Simons theory. For $N_f = 1$, the only interface must end with a boundary in the interior. For $N_f = 2$, the two interfaces can connect into one interface. For $N_f > 2$, the interfaces can form junctions at the origin. The $SU(N)_{-1}$ Chern-Simons theories on each interfaces provide a gravitational anomaly inflow to the interior. The gravitational anomaly inflow is determined by the framing anomaly or the chiral central charge $c$ of the Chern-Simons theory [35, 36]. It is characterized by the anomaly polynomial

$$- 4\pi c \frac{\mathrm{Tr}(R \wedge R)}{384\pi^2} \ . \tag{3.3}$$

The chiral central charge of the $SU(N)_{-1}$ theory is $c = 1 - N$.

Combining the contributions from the winding counterterm and from the $N_f$ $SU(N)_{-1}$ Chern-Simons theories, the total anomaly inflow to the interior is

$$- 2\pi N N_f \frac{F_A \wedge F_A}{8\pi^2} - 2\pi N \frac{\mathrm{Tr}(F_B \wedge F_B)}{8\pi^2} - 4\pi N_f \frac{\mathrm{Tr}(R \wedge R)}{384\pi^2} \ . \tag{3.4}$$

The anomaly inflow only constrains the dynamics in the interior. Below, we will present a coherent picture of the interior using various low energy effective field theories for the bulk and for the interfaces.

## 4   Interior for $N_f = 1$

Here we focus on QCD with one quark. The analysis in section 3 shows that at large radius, the theory has only one interface centered at $\alpha = \pi$ with an $SU(N)_{-1}$ Chern-Simons theory on its worldvolume together with a winding counterterm

$$N\alpha \frac{F_A \wedge F_A}{8\pi^2} + 2N\alpha \frac{\mathrm{Tr}(R \wedge R)}{384\pi^2} \ . \tag{4.1}$$

We can deform the counterterm such that it jumps across the interface

$$2\pi N \Theta(\alpha - \pi) \left( \frac{F_A \wedge F_A}{8\pi^2} + 2 \frac{\mathrm{Tr}(R \wedge R)}{384\pi^2} \right) \ . \tag{4.2}$$

Here $\Theta(x)$ is the Heaviside step function. This generates some classical Chern-Simons terms on the interface

$$- \frac{N}{4\pi} A dA - 2N \mathrm{CS}_{\mathrm{grav}} \ . \tag{4.3}$$

Here $A$ is a $U(1)$ gauge field with $2\pi$ integer flux. More generally, we can promote $A$ to a $U(1)/\mathbb{Z}_N$ gauge field with fractional flux since the faithful global symmetry is $U(1)/\mathbb{Z}_N$. This twists the dynamical gauge bundle to a $PSU(N)$ bundle such that the combination $\widetilde{c} = c + A\mathbb{1}$ is always a well-defined $U(N)$ gauge field. This allows us to combine the dynamical $SU(N)_{-1}$ Chern-Simons theory with the classical Chern-Simons term (4.3), which gives

$$\mathcal{L} = -\frac{1}{4\pi}\text{Tr}\left(\widetilde{c}d\widetilde{c} - \frac{2i}{3}\widetilde{c}^3\right) - \frac{1}{2\pi}ad(\text{Tr}(\widetilde{c}) - NA) - 2N\text{CS}_{\text{grav}}\ , \tag{4.4}$$

where $a$ is a dynamical $U(1)$ gauge field that acts as a Lagrangian multiplier. The combined theory is level-rank dual to a $U(1)_N$ Chern-Simons theory with no additional counterterms [37]:

$$\mathcal{L} = \frac{N}{4\pi}ada + \frac{N}{2\pi}adA\ . \tag{4.5}$$

The background gauge field $A$ now couples to the magnetic $U(1)$ symmetry of the dual theory. We will use the $U(1)_N$ description for the interface theory below.

The theory has only one interface at large radius so the interface has to terminate somewhere in the bulk. The bulk phase diagram suggests that the interface continues all the way along $\alpha = \pi$ following the first-order phase transition line and ends around $x_0 = M_0/(\varepsilon\Lambda^2)$ where the quadratic potential of the $\eta'$ particle vanishes. The dynamics in the interior is summarized in figure 2.

On the interface, at certain radius $R_1 \sim 1/(\varepsilon\Lambda)$ where $|m| \sim \Lambda$, the theory undergoes a transition from a $U(1)_N$ Chern-Simons theory to a trivial theory. We will assume the transition can be described by an effective theory on the interface: a $U(1)_N$ Chern-Simons theory coupled to a complex scalar. The transition leads to an interface in the effective theory where the mass squared of the scalar interpolates from a positive value to a negative value. We will refer to this interface as the transition interface. The theory is equivalent to a $U(1)_N$ Chern-Simons theory on one side where the scalar is massive, and a trivial theory on the other side where the scalar has a non-zero vacuum expectation value.

The transition interface is analyzed in appendix A. We will use the light coordinate for the transverse directions

$$x_{\pm} = \frac{1}{2}(t \pm z), \quad \partial_{\pm} = \partial_t \pm \partial_z, \quad a_{\pm} = a_t \pm a_z\ . \tag{4.6}$$

It is shown that the transition leads to a chiral Dirichlet boundary condition $a_- = 0$ on the transition interface, which gives rise to a 1+1 dimensional compact chiral boson on the interface [38–40]. The action of the chiral boson is

$$S = \int d^2x \left(-\frac{N}{4\pi}\partial_-\phi\partial_+\phi + \frac{N}{2\pi}A_-\partial_+\phi\right)\ . \tag{4.7}$$

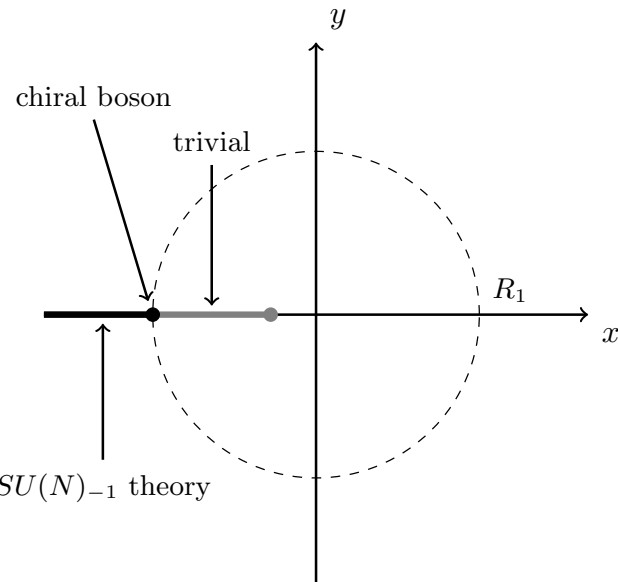

Figure 2: The dynamics in the interior for a space-dependent mass profile $m = \varepsilon r e^{i\alpha} \Lambda^2$ when $N_f = 1$. The interface extends along the negative $x$ axis and terminates at some point. At radius $R_1$, the theory on the interface undergoes a transition from an $SU(N)_{-1}$ Chern-Simons theory to a trivially gapped theory which leads to a 1+1 dimensional compact chiral boson on the transition interface.

$\phi$ is periodic $\phi \sim \phi + 2\pi$, and it has a gauge symmetry $\phi \to \phi + \lambda(x_-)$ which eliminates the anti-chiral modes. The background gauge field $A$ for the baryon symmetry now couples to the shift symmetry of the compact boson. Under the background gauge transformation,

$$A \to A + d\xi, \quad \phi \to \phi + \xi , \tag{4.8}$$

the action is not invariant instead it is shifted by

$$S \to S + \int d^2x \left( \frac{N}{4\pi} \partial_- \xi \partial_+ \xi + \frac{N}{2\pi} A_- \partial_+ \xi \right) . \tag{4.9}$$

This represents an 't Hooft anomaly that can be canceled by the following 2+1 dimensional invertible field theory together with a 1+1 dimensional counterterm

$$- \frac{N}{4\pi} \int d^3x\, \epsilon^{\mu\nu\rho} A_\mu \partial_\nu A_\rho - \frac{N}{4\pi} \int d^2x\, A_- A_+ . \tag{4.10}$$

The corresponding anomaly polynomial matches with the $U(1)$ part of the anomaly inflow (3.4) from infinity. The chiral central charge of the theory $c_L - c_R = 1$ also agrees with the anomaly inflow.

Interestingly, the excitations of the chiral boson can carry baryon charges. Consider a vertex operator $\mathcal{O}_N = e^{iN\phi}$. The operator carries $N$ $U(1)$ charge or equivalently one unit of baryon charge. The corresponding excitation can be interpreted as baryons in the ultraviolet theory, $\epsilon^{a_1 \cdots a_N} \psi_{a_1} \cdots \psi_{a_N}$, that are localized on the 1+1 dimensional transition interface. This observation has been recently employed in the quantum Hall droplet proposal for the $N_f = 1$ baryons [15]. Let us briefly summarize the proposal. Within the large $N$ effective field theory of $\eta'$, one can define a two-form global symmetry associated to the topological current

$$J_{\mu\nu\rho} = \frac{1}{2\pi} \epsilon_{\mu\nu\rho\sigma} \partial^\sigma \eta' . \tag{4.11}$$

The charged objects are two-dimensional sheets. Such two-form symmetry is absent in the ultraviolet theory, which means that the charged sheets are only meta-stable excitations in the full theory. It was proposed that these meta-stable sheets have a compact chiral boson on their boundary, and following the same discussions above, the sheets with nontrivial boundary excitations can be interpreted as baryons in the ultraviolet theory [15].[9] As a nontrivial check, the operator $\mathcal{O}_N$ has spin $N/2$, which precisely matches with the spin of the baryons. The chiral boson has more basic vertex operators such as $\mathcal{O}_1 = e^{i\phi}$ which can be thought as the end point of the charge 1 Wilson line in the $U(1)_N$ Chern-Simons theory. This operator should be contrasted with the genuine local operator $\mathcal{O}_N = e^{iN\phi}$

---

[9]See [41] for discussions on the connections between the $N_f = 1$ quantum Hall droplet and the $N_f > 1$ Skyrmions.

that connects to a transparent Wilson line in the Chern-Simon theory. The excitations corresponding to $\mathcal{O}_1$ have fractional $1/N$ baryon charge. Hence they can be interpreted as a liberated quark localized on the transition interface.

Let us come back to our analysis of the interface. Around the end point of the interface, the effective field theory of the $\eta'$ particle is valid. Its Lagrangian is

$$\mathcal{L} = f_{\eta'}^2 \left( \frac{1}{2}(\partial \eta')^2 + \kappa(y)\eta' + \frac{1}{2}\mu^2(x)\eta'^2 + \frac{1}{4}\lambda \eta'^4 \right) , \tag{4.12}$$

where $\kappa(y) \propto \varepsilon y$ and $\mu^2(x) \propto \varepsilon(x - x_0)$. The quadratic term in the $\eta'$ potential vanishes at $x_0$. One may suspect that the $\eta'$ particle gets localized at $x_0$ and becomes gapless 1+1 dimensional excitations. In appendix B, we analyze the system in an extreme scenerio where $\kappa(y)$ and $\mu^2(x)$ are discontinuous step functions. We observe that the 1+1 dimensional effective mass of the $\eta'$ particle is always positive. Even though the analysis is restricted to discontinuous profiles, we expect the conclusion holds for generic profiles. This suggests that there are no gapless degrees of freedom localized at $x_0$.

Let us compare the $\eta'$ profile in our system with the one around the $\eta'$ sheet. We will work with the large $N$ chiral Lagrangian (2.3). Consider the theory at a radius which is large compared to the scales in (4.12) but sill small enough such that the chiral Lagrangian is valid. At such radius, the potential energy dominates the chiral Lagrangian so in order to minimize the potential, $\eta' \approx -\alpha$ for $\alpha \in (-\pi + \epsilon, \pi - \epsilon)$ where $\epsilon$ denotes a small angle. In order to avoid the singularity at the origin, $\eta'$ can not have a nontrivial winding number at large radius so it increases rapidly from $\eta' \approx -\pi + \epsilon$ to $\eta' \approx \pi - \epsilon$ for $\alpha \in (\pi - \epsilon, \pi + \epsilon)$. This leads to an interface centered at $\alpha = \pi$. The $\eta'$ profile is everywhere smooth and never crosses $\eta' = \pi$. It is to be contrasted with the $\eta'$ sheet where $\eta'$ changes by $2\pi$ across the sheet and it remains constant away from the sheet. This configuration has a nontrivial winding around the boundary of the sheet, and hence generates a singularity on the boundary. The singularity can not be computed in the $\eta'$ effective field theory but it can be resolved in the full theory by a vanishing vacuum expectation value of the chiral condensate.

## 5  Interior for $N_f > 1$

We now consider QCD with more than one quark. The $N_f = 2$ case will be discussed separately in a subsection. We will assume $N_{\mathrm{CFT}} > N_f$ so that the chiral Lagrangian is valid. The analysis in section 3 shows that at large radius, the theory has $N_f$ interfaces centered at $\alpha = (\pi + 2\pi\mathbb{Z})/N_f$ with an $SU(N)_{-1}$ Chern-Simons theory on the worldvolume

of each interface. There is also a winding counterterm

$$NN_f\alpha\frac{F_A \wedge F_A}{8\pi^2} + N\alpha\frac{\text{Tr}(F_B \wedge F_B)}{8\pi^2} + 2NN_f\alpha\frac{\text{Tr}(R \wedge R)}{384\pi^2} \ . \tag{5.1}$$

As in the $N_f = 1$ case, we can deform the counterterm such that the gravitational coun-
terterm and the $U(1)$ counterterm jump across each interface

$$N\alpha\frac{\text{Tr}(F_B \wedge F_B)}{8\pi^2} + 2\pi N\sum_{a=1}^{N_f}\Theta\left(\alpha - \frac{(2a-1)\pi}{N_f}\right)\left(\frac{F_A \wedge F_A}{8\pi^2} + 2\frac{\text{Tr}(R \wedge R)}{384\pi^2}\right) \ . \tag{5.2}$$

This generates the classical Chern-Simons term (4.3) on each interface. With this coun-
terterm, the interface theory can be dualized to a $U(1)_N$ Chern-Simons theory. There still
remains a winding counterterm for the $SU(N_f)$ gauge field $B$.

The dynamics in the interior depends on how large $\varepsilon$ is. The discussions will be divided
into two cases: $\varepsilon \ll 1$ and $\varepsilon \gtrsim 1$.

## 5.1   Slowly Varying Mass Profile

We will first consider the case of $\varepsilon \ll 1$. The dynamics in the interior is summarized in
figure 3. The interfaces at large radius extend towards the interior along the radial direction.
When they enter certain radius $R_1 \sim 1/(\varepsilon\Lambda)$ where $|m| \sim \Lambda$, the theories on the interfaces
undergo a transition from a $U(1)_N$ Chern-Simons theory to a $\mathbb{CP}^{N_f-1}$ sigma model with a
Wess-Zumino term. At the transition radius $R_1$, $|\nabla\theta| \sim 1/R_1 \sim \varepsilon\Lambda \ll \Lambda$, the interfaces are
still far apart.

In general, there are many different boundary conditions one can choose between the
$U(1)_N$ Chern-Simons theory and the $\mathbb{CP}^{N_f-1}$ sigma model. Here we determine the boundary
condition by assuming that the transition is described by an effective theory on the interface:
a $U(1)_N$ Chern-Simons theory coupled to $N_f$ complex scalars. The transition leads to an
interface in the effective theory where the mass squared of the scalars interpolates from a
positive value to a negative value. We will refer to this interface as the transition interface.
On one side of the transition interface, the scalars are massive so the theory is equivalent
to a $U(1)_N$ Chern-Simons theory. On the other side, the scalars have a non-zero vacuum
expectation value so the theory is equivalent to a $\mathbb{CP}^{N_f-1}$ sigma model.

As discussed in section 2.2, the $\mathbb{CP}^{N_f-1}$ sigma model can be parameterized by $N_f$ com-
plex scalars $\Phi_I$ with a constraint $\sum\Phi_I^\dagger\Phi_I = 1$ and a $U(1)$ gauge symmetry $\Phi_I \to \Phi_I e^{-i\lambda}$.
We define a composite gauge field $b = i\sum\Phi_I^\dagger d\Phi_I$ that transforms as an ordinary gauge field
under the gauge symetry.

The transition interface is analyzed in appendix A. It is shown that the transition leads

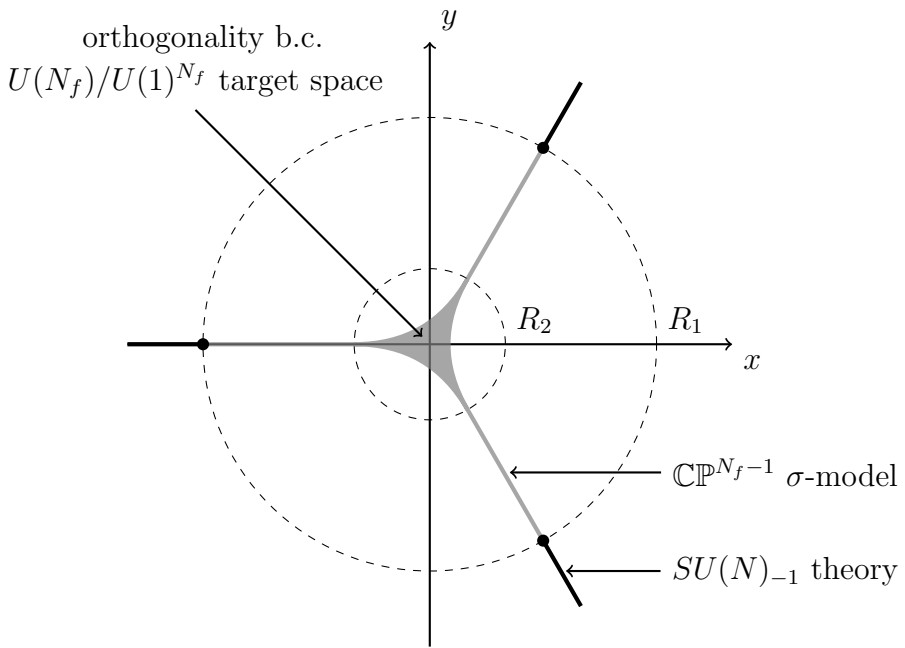

Figure 3: The dynamics in the interior for a space-dependent mass $m = \varepsilon r e^{i\alpha}\Lambda^2$ with $\varepsilon \ll 1$ when $N_f = 3$. There are $N_f$ interfaces continuing from infinity to the origin. When the interfaces pass through radius $R_1 \sim 1/(\varepsilon\Lambda)$, the theories on the interfaces undergo a transition from $SU(N)_{-1}$ Chern-Simons theory to a $\mathbb{CP}^{N_f-1}$ sigma model with a Wess-Zumino term which leads to 1+1 dimensional chiral modes localized at the transition interfaces. The interfaces form an interface junction of size $R_2 \sim 1/(\varepsilon^{1/3}\Lambda)$ at the origin. On the interface junction, one should impose an orthogonality boundary condition $\sum_I \Phi_{Ia}^{\dagger}\Phi_{Ib} = \delta_{ab}$ for the $N_f$ $\mathbb{CP}^{N_f-1}$ sigma models. This reduces the target space on the junction from $N_f$ independent copies of $\mathbb{CP}^{N_f-1}$ manifold to the flag manifold $U(N_f)/U(1)^{N_f}$. The flag sigma model on the junction is supplemented by a Wess-Zumino term $S_{\mathrm{WZW}}$ defined in (5.22).

to a chiral boundary condition $a_- = b_-$ that identifies a light cone component of the Chern-Simons gauge field $a$ with the corresponding light cone component of the composite gauge field $b$ on the interface. The $U(1)$ gauge symmetries on both sides share the same gauge parameter on the interface. As discussed in appendix A, after integrating out the Chern-Simons gauge field and fixing the gauge symmetry, a boundary term on the transition interface is generated for the $\mathbb{CP}^{N_f-1}$ sigma model. The action of the system is

$$S = \int d^3x \left( v_0^2 D^\mu \Phi_I^\dagger D_\mu \Phi_I + \frac{N}{4\pi} \epsilon^{\mu\nu\rho} b_\mu \partial_\nu b_\rho \right) - \frac{N}{4\pi} \int d^2x \, b_- b_+ \,, \tag{5.3}$$

where $D\Phi_I = (\partial + ib)\Phi_I$. Any two-dimensional integral should be understood as the integral on the transition interface. The $U(1)$ gauge symmetry of $\Phi_I$ is restricted on the transition interface such that the gauge parameter can depends on $x_-$ on the boundary. This leads to a chiral mode that depends only on $x_+$ which absorbs the gravitational anomaly inflow (3.4) from infinity. We can couple the system to the $U(1)$ background gauge field $A$ through

$$\frac{N}{2\pi} \int d^3x \, \epsilon^{\mu\nu\rho} A_\mu \partial_\nu b_\rho - \frac{N}{2\pi} \int d^2x \, A_- b_+ \,. \tag{5.4}$$

The background $A$ couples to both the skyrmion current in the bulk of the interface and the current localized on the transition interface. These currents should be interpreted as baryon currents. Under the background gauge transformation

$$A \to A + d\xi, \quad \Phi_I \to e^{i\xi} \Phi_I, \quad b \to b - d\xi \,, \tag{5.5}$$

the action is not invariant instead it is shifted by

$$S \to S + \int d^2x \left( \frac{N}{4\pi} \partial_- \xi \partial_+ \xi + \frac{N}{2\pi} A_- \partial_+ \xi \right) \,. \tag{5.6}$$

This represents an 't Hooft anomaly that can be canceled by the 2+1 dimensional invertible field theory (4.10). The corresponding anomaly polynomial matches with the $U(1)$ part of the anomaly inflow (3.4) from infinity.

After passing through the transition radius $R_1$, the interfaces continue towards the origin. In the domain around the origin where $|m| \ll \Lambda$, the low energy dynamics can be described by the chiral Lagrangian with a space-dependent potential

$$\mathcal{L}_\pi = \frac{1}{2} f_\pi^2 \text{Tr}(\partial_\mu U^\dagger \partial^\mu U) - \frac{1}{2} \varepsilon f_\pi^2 \Lambda^3 \left( r e^{i\alpha} \text{Tr}(U) + r e^{-i\alpha} \text{Tr}(U^\dagger) \right) \,. \tag{5.7}$$

The interfaces are still far apart whenever the condition $|\nabla\theta| \sim 1/r \ll \sqrt{m\Lambda} = \sqrt{\varepsilon r \Lambda^3}$ holds. However, the condition is violated within the radius $R_2 \sim 1/(\varepsilon^{1/3}\Lambda)$ where the

interfaces smear out and form an interface junction (see figure 3).

Between $R_1$ and $R_2$, each interface supports a $\mathbb{CP}^{N_f-1}$ sigma model. As discussed in section 2.2, these sigma models arise because the eigenvalues of the bulk field $U$ interpolate across the interfaces following different trajectories which break the $SU(N_f)$ symmetry of the bulk chiral Lagrangian.

Now with a winding mass the eigenvalues will depend on both the radial and the angular coordinates. Let us denote the vacuum eigenvalue matrix of the bulk field $U$ by

$$
V(r,\alpha) = \begin{pmatrix} e^{i\varphi_1} & & & \\ & e^{i\varphi_2} & & \\ & & \ddots & \\ & & & e^{i\varphi_{N_f}} \end{pmatrix}, \quad \sum \varphi_a = 0 \bmod 2\pi \ . \tag{5.8}
$$

All vacua can be generated by the $SU(N_f)$ symmetry $V \to gVg^\dagger$. Outside the radius $R_2$, the eigenvalues depend only on the angular coordinate $\alpha$. We will determine how the eigenvalues interpolate as $\alpha$ winds outside $R_2$. Let us partition the angular coordinate $\alpha$ into $N_f$ intervals labeled by $n = 0, \cdots, N_f - 1$

$$
I_n = [\alpha_n, \alpha_{n+1}], \qquad \alpha_n = \frac{2\pi n}{N_f}. \tag{5.9}
$$

Within each interval, there is exactly one interface so the eigenvalues should follow the same trajectories as how they interpolate across an interface described in section 2.2. On the $n$-th interval $I_n$, the phases of the eigenvalues interpolate from $(-\alpha_n \bmod 2\pi)$ to $(-\alpha_{n+1} \bmod 2\pi)$. $N_f - 1$ of them interpolate using $\varphi(\alpha - \alpha_n)$ and the remaining one interpolates in the opposite direction using $-(N_f - 1)\varphi(\alpha - \alpha_n)$. Here $\varphi(\alpha)$ is a function on $[0, 2\pi/N_f]$ which interpolates from 0 to $-2\pi/N_f$. It is crucial that these phases can not have nontrivial winding numbers at infinity, otherwise, they develop a singularity in the interior. The only way to achieve this is to demand that each phase interpolates using $-(N_f - 1)\varphi(\alpha - \alpha_n)$ once in one of the $N_f$ intervals. Gluing the $N_f$ intervals together, the phases are smooth functions of $\alpha$ in the following form

$$
\varphi_a = -N_f \varphi\left(\alpha - \alpha_a\right) \Pi_a + \sum_{n=a+1}^{N_f} 2\pi \Pi_n + \sum_{n=0}^{N_f-1} \left(\varphi\left(\alpha - \alpha_n\right) - \alpha_n\right) \Pi_{n+1} \ . \tag{5.10}
$$

Here $\Pi_n(\alpha) = \Theta(\alpha - \alpha_{n-1}) - \Theta(\alpha - \alpha_n)$ denotes a rectangular function that has support only on the interval $I_{n-1}$. Figure 4 sketches how the phases interpolate as $\alpha$ winds outside $R_2$. They set the boundary conditions for the phases inside $R_2$.

Different eigenvalues have different boundary conditions at the radius $R_2$ so the $SU(N_f)$ global symmetry is broken to its Cartan subgroup $U(1)^{N_f-1}$ inside $R_2$. This leads to a 1+1

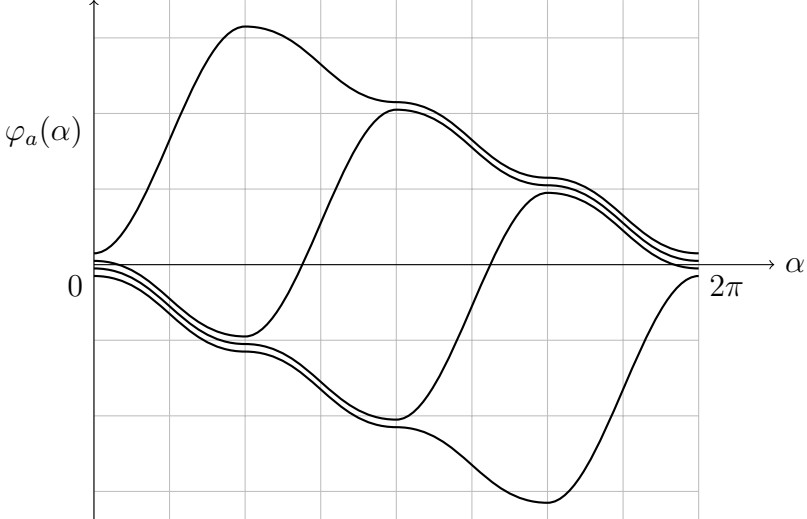

Figure 4: The trajectories of the phases $\varphi_a(\alpha)$ outside $R_2$ for $N_f = 4$. We deliberately split the trajectories by a tiny amount for a better illustration. The trajectories that are close to each other suppose to coincide.

dimensional sigma model at the center whose target space is the flag manifold

$$\frac{U(N_f)}{\prod_{a=1}^{N_f} U(1)} . \tag{5.11}$$

We will refer to this sigma model as flag sigma model.[10] We emphasize that the flag sigma model at the center is not an isolated theory. It lives on the junction of the $N_f$ incoming interfaces so it should be viewed as the boundary theory of the $N_f$ copies of gapless 2+1 dimensional $\mathbb{CP}^{N_f-1}$ sigma model on the interfaces. As we will explain below, the flag sigma model does not have any additional degrees of freedom compared to the $N_f$ $\mathbb{CP}^{N_f-1}$ sigma models. All of its fields come from restricting the fields of the $N_f$ $\mathbb{CP}^{N_f-1}$ sigma models to the junction. Since the flag sigma model lives on the boundary of 2+1 dimensional theories, its action can include terms that do not exist in the isolated theory. For instance, as we will see later, the action of the flag sigma model includes a gauge non-invariant Wess-Zumino term $S_{\mathrm{WZW}}$ defined in (5.22) whose gauge non-invariance is canceled by the gauge variations of the three-dimensional theories.

Let us parametrize the $\mathbb{CP}^{N_f-1}$ sigma models on the $N_f$ interfaces by an $N_f \times N_f$ matrix $\Phi_{Ia}$ where $a$ labels the $N_f$ interfaces and $I$ labels the $N_f$ complex scalar fields on each interface. The scalar fields have a constraint $\sum_I \Phi_{Ia}^\dagger \Phi_{Ia} = 1$ and a $U(1)$ gauge symmetry $\Phi_{Ia} \to \Phi_{Ia} e^{-i\lambda_a}$ for each $a$. We comment that $\Phi_{Ia}$ is not a $U(N_f)$ matrix in general.

---

[10]See [42–44] for discussions on 1+1 dimensional sigma models whose target space is a flag manifold.

We can also parametrize each sigma model by a $U(N_f)$ matrix with a block-diagonal $U(1) \times U(N_f - 1)$ gauge symmetry acting from the right. The matrix on the $a$-th interface is denoted by $h^a$. Since the trajectory of the $a$-th eigenvalue is different from the others on the $a$-th interface, we will demand that that the $U(1)$ gauge symmetry of $h^a$ acts on its $a$-th column vector. The two parametrization are related by $(h^a)_{Ia} = \Phi_{Ia}$.

The flag manifold can be parametrized by a $U(N_f)$ matrix $g_{Ia}$ with a block-diagonal $\prod_{a=1}^{N_f} U(1)$ gauge symmetry acting from the right $g_{Ia} \to g_{Ia} e^{-i\lambda_a}$. The unitarity condition imposes a constraint $\sum g_{Ia}^\dagger g_{Ib} = \delta_{ab}$.

Since the flag sigma model and the $\mathbb{CP}^{N_f-1}$ sigma models all arise from the symmetry breaking of the bulk theory, we should be able to relate the two target spaces. The $N_f$ copies of $\mathbb{CP}^{N_f-1}$ manifold can be embedded in the flag manifold through the identification $g_{Ia} = \Phi_{Ia}$. The constraint and the gauge symmetry of each $\mathbb{CP}^{N_f-1}$ sigma model is included in the unitarity constraint and the gauge symmetries of the flag sigma model. On the other hand, the flag sigma model is more constrained than $N_f$ independent copies of $\mathbb{CP}^{N_f-1}$ sigma model. The flag sigma model requires that the $N_f$ scalar fields from different copies are orthogonal,

$$\sum \Phi_{Ia}^\dagger \Phi_{Ib} = 0 \quad \text{for } a \neq b . \tag{5.12}$$

The orthogonality condition should be viewed as a boundary condition at the junction for the $N_f$ copies of $\mathbb{CP}^{N_f-1}$ sigma model on the interfaces. The boundary condition reduces the target space from $N_f$ copies of $\mathbb{CP}^{N_f-1}$ manifold to the flag manifold which becomes the target space of the 1+1 dimensional sigma model on the junction.

Let us determine the effective action of the flag sigma model on the junction. Specifically, we will consider the reduction of the Wess-Zumino term $N\Gamma_{\text{WZ}}$ of the bulk chiral Lagrangian.

The bulk field $U$ is related to the fields on the interfaces and the fields on the interface junction through

$$U(r, \alpha, \vec{z}) = \begin{cases} g(\vec{z})V(r, \alpha)g(\vec{z})^\dagger, & \text{for } r \leq R_2 \\ \sum_{a=1}^{N_f} \Pi_a(\alpha)h^a(r, \vec{z})V(\alpha)h^a(r, \vec{z})^\dagger, & \text{for } r \geq R_2 \end{cases} . \tag{5.13}$$

It is invariant under the gauge symmetries of $g$ and $h^a$. It is also continuous everywhere, especially at radius $R_2$ thanks to the identification $g_{Ia} = (h^a)_{Ia}$. To compute the Wess-Zumino term, we construct a five-dimensional extension of $U$ by extending $g(\vec{z})$ to a three-dimensional manifold $N_3$ and $h^a(r, \vec{z})$ to a four-dimensional manifold $N_4^a$ for each interface. The boundary of $N_3$ is the worldsheet of the junction $M_2$ and the boundary of $N_4^a$ includes $N_3$ and the worldvolume of the $a$-th interface $M_3^a$. The extension of $g_{Ia}$ and $(h^a)_{Ia}$ should agree on $N_3$. The details of the extension are illustrated in figure 5.

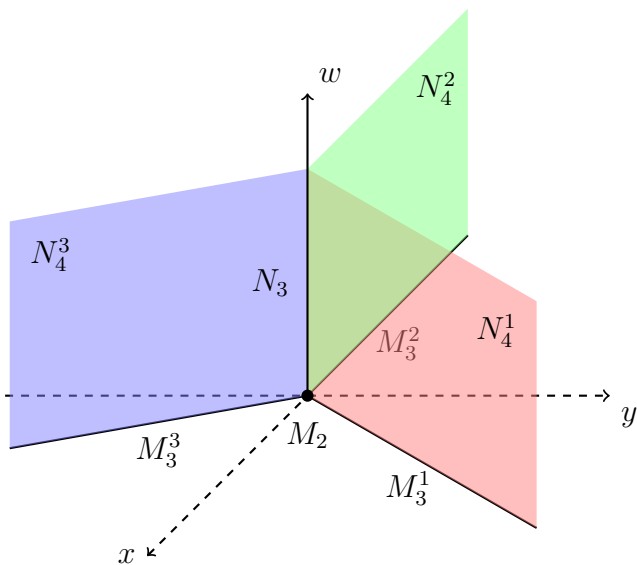

Figure 5: The geometry of the extensions of the interfaces and the interface junction. The two transverse directions are not shown in the figure. $w$ is the coordinate for the fifth-dimension.

We notice that

$$g^\dagger U^\dagger dU g = V^\dagger g^\dagger dg V + V^\dagger dV - g^\dagger dg \ . \tag{5.14}$$

Outside $R_2$, $V$ depends only on the angular coordinate so we need exactly one factor of $V^\dagger dV$. As computed in section 2.2, this gives

$$\sum_a \frac{N}{4\pi} \int_{N_4^a} db^a db^a = \sum_a \frac{N}{4\pi} \int_{M_3^a} b^a db^a - \frac{N}{4\pi} \int_{N_3} \sum_a b^a db^a \ , \tag{5.15}$$

where we define $b^a = i \sum_I \Phi^\dagger_{Ia} d\Phi_{Ia}$. The minus sign of the second term comes from the orientation of $N_3$ which is chosen such that $\partial N_3 = \partial M_3^a = M_2$. Inside $R_2$, $V$ depends only on the radial and the angular coordinates so we need exactly two factors of $V^\dagger dV$. This gives

$$\frac{iN}{48\pi^2} \left( \int_{N_3 \times D^2} \mathrm{Tr} \left[ (V^\dagger dV)^2 (V^\dagger g^\dagger dg V - g^\dagger dg)^3 \right] + \right.$$
$$\left. \int_{N_3 \times D^2} \mathrm{Tr} \left[ (V^\dagger dV)(V^\dagger g^\dagger dg V - g^\dagger dg)(V^\dagger dV)(V^\dagger g^\dagger dg V - g^\dagger dg)^2 \right] \right) \ . \tag{5.16}$$

The first term vanishes since $(V^\dagger dV)^2 = 0$. The second term simplifies to

$$\frac{N}{24\pi^2} \int_{N_3} C_{abc} (g^\dagger dg)_{ab} (g^\dagger dg)_{bc} (g^\dagger dg)_{ca} . \tag{5.17}$$

The coefficient $C_{abc}$ is determined by the following integral

$$\begin{aligned}
C_{abc} &= -\frac{1}{3} \int_{r<R_2} (\sin(\varphi_{ab}) + \sin(\varphi_{bc}) + \sin(\varphi_{ca})) (d\varphi_a d\varphi_b + d\varphi_b d\varphi_c + d\varphi_c d\varphi_a) \\
&= -\frac{1}{6} \int_{r=R_2} \cos(\varphi_{ab}) d(\varphi_{ca} + \varphi_{cb}) + \cos(\varphi_{bc}) d(\varphi_{ab} + \varphi_{ac}) + \cos(\varphi_{ca}) d(\varphi_{bc} + \varphi_{ba})
\end{aligned} \tag{5.18}$$

where we rewrite the integral into a boundary integral using the Stokes' theorem. On the boundary, the phases are in the form of (5.10). The integrand can be expanded as follows

$$\begin{aligned}
d\varphi_{ab} &= N_f (\Pi_b - \Pi_a) d\varphi , \\
\cos(\varphi_{ab}) &= 1 + (1 - \delta_{ab})(\cos(N_f \varphi) - 1)(\Pi_a + \Pi_b) ,
\end{aligned} \tag{5.19}$$

using which, the integral evaluates to

$$C_{abc} = -2\pi(1 - \delta_{ab})(1 - \delta_{bc})(1 - \delta_{ca}) . \tag{5.20}$$

In the end, inside $R_2$, the Wess-Zumino term reduces to

$$-\frac{N}{12\pi} \int_{N_3} \mathrm{Tr}\left[(g^\dagger dg)^3\right] + \frac{N}{4\pi} \int_{N_3} \sum_a b^a db^a , \tag{5.21}$$

where we use the identification $g_{Ia} = \Phi_{Ia}$ on $N_3$ to replace $i(g^\dagger dg)_{aa}$ by $b^a$.

Combining the two terms (5.15) and (5.21), we conclude that the Wess-Zumino term $N\Gamma_{\mathrm{WZ}}$ reduces to

$$\underbrace{-\frac{N}{12\pi} \int_{N_3} \mathrm{Tr}\left[(g^\dagger dg)^3\right]}_{S_{\mathrm{WZW}}} + \frac{N}{4\pi} \sum_a \int_{M_3^a} b^a db^a . \tag{5.22}$$

The first term $S_{\mathrm{WZW}}$ is the Wess-Zumino term of the $U(N_f)$ field $g$ which does not depend on the extensions. It is not gauge invariant under the $\prod_{a=1}^{N_f} U(1)$ gauge symmetry. The second term is the Wess-Zumino term of the $\mathbb{CP}^{N_f-1}$ sigma models on the interfaces. This Wess-Zumino term is also not gauge invariant under the gauge symmetry but its variation can be written as a boundary term which cancels the gauge variation of $S_{\mathrm{WZW}}$ on the junction.

The $SU(N_f)$ global symmetry acts on the flag sigma model as $g \to hg$. Let us couple the symmetry to the $SU(N_f)$ background gauge field $B$. This modifies the Wess-Zumino

term $S_{\text{WZW}}$ to

$$S_{\text{GWZW}} = -\frac{N}{12\pi} \int_{N_3} \text{Tr}\left[(g^\dagger dg)^3\right] - \frac{N}{4\pi} \int_{M_2} \text{Tr}(iBdgg^\dagger) \ . \tag{5.23}$$

$S_{\text{GWZW}}$ is not invariant under the infinitesimal background gauge transformation

$$B \to B + [B, \zeta] + d\zeta, \quad g \to g + i\zeta g \ . \tag{5.24}$$

It transforms as

$$S_{\text{GWZW}} \to S_{\text{GWZW}} + \frac{N}{4\pi}\text{Tr}(Bd\zeta) \tag{5.25}$$

This represents an 't Hooft anomaly that can be canceled by the following $2+1$ dimensional invertible field theory

$$-\frac{N}{4\pi} \int d^3x \, \text{Tr}\left(BdB - \frac{2i}{3}B^3\right) \tag{5.26}$$

which matches the $SU(N_f)$ part of the anomaly inflow (3.4) from infinity. The excitations of the flag sigma model does not carry any baryon charges. It can be seen directly by reducing the skyrmion current (2.6) to the junction.

The dynamics in the interior when $\varepsilon \ll 1$ is summarized in figure 3. There are $N_f$ interfaces continuing from infinity to the origin. When the interfaces pass through radius $R_1$, the theories on the interfaces undergo a transition from $SU(N)_{-1}$ Chern-Simons theory to a $\mathbb{CP}^{N_f-1}$ sigma model which leads to $1+1$ dimensional chiral modes localized at the transition interfaces. The interfaces form an interface junction of size $R_2$ at the origin. On the interface junction, one should impose an orthogonality boundary condition $\sum_I \Phi_{Ia}^\dagger \Phi_{Ib} = \delta_{ab}$ for the $N_f$ $\mathbb{CP}^{N_f-1}$ sigma models. This reduces the target space on the junction to the flag manifold $U(N_f)/U(1)^{N_f}$. The flag sigma model on the junction is supplemented by a Wess-Zumino term $S_{\text{WZW}}$.

## 5.2 Rapidly Varying Mass Profile

We now increase $\varepsilon$. It shrinks the domain between $R_1$ and $R_2$ where the interface supports a gapless $\mathbb{CP}^{N_f-1}$ sigma model. When $\varepsilon$ is order 1, $R_1 \sim R_2 \sim 1/\Lambda$ so the theory on the junction is in direct contact with the gapped topological field theories on the interfaces. Since the interfaces are gapped, the gapless degrees of freedom on the junction can be described alone by a $1+1$ dimensional theory. We will use the $U(1)_N$ description for the topological field theories on the interfaces.

When $\varepsilon \ll 1$, the boundary condition at the transition interface on the $a$-th interface is

$$a_-^a = i\sum_I \Phi_{Ia}^\dagger \partial_- \Phi_{Ia} \ . \tag{5.27}$$

On the junction, the field $g_{Ia}$ of the flag sigma model are identified with the field $\Phi_{Ia}$ of the $\mathbb{CP}^{N_f-1}$ sigma models. When $\varepsilon$ is order 1, the transition interfaces collapse with the junction and the two boundary conditions combine into

$$a^a_- = i(g^\dagger \partial_- g)_{aa} \ , \tag{5.28}$$

on the junction. We will use the same notation $b^a$ to denote $i(g^\dagger dg)_{aa}$.

The boundary condition can be imposed dynamically by equations of motion. As discussed in appendix A.1, this amounts to introducing a boundary term on the junction. The action of the system is modified to

$$S = +\frac{N}{4\pi}\sum_a \int_{M_3^a} a^a da^a$$
$$-\frac{N}{4\pi}\int_{M_2} d^2x \left(\text{Tr}(\partial_- g^\dagger \partial_+ g) - 2\sum_a a^a_+ b^a_- + \sum_a a^a_- a^a_+\right) - \frac{N}{12\pi}\int_{N_3} \text{Tr}\left[(g^\dagger dg)^3\right] \ . \tag{5.29}$$

It is invariant under the $\prod_{a=1}^{N_f} U(1)$ gauge symmetry. The boundary condition $a^a_- = b^a_-$ is imposed by the equation of motion of $a$ on the junction. Similar to the discussions in appendix A, we can integrate out $a^a_-$ in the Chern-Simons theory and solve the constraint by introducing compact bosons $\phi^a$

$$a^a_+ = \partial_+ \phi^a, \quad a^a_x = \partial_x \phi^a \ . \tag{5.30}$$

The action then becomes

$$S = -\frac{N}{4\pi}\int_{M_2} d^2x \left(\text{Tr}(\partial_- g^\dagger \partial_+ g) - 2\sum_a \partial_+ \phi^a b^a_- + \sum_a \partial_- \phi^a \partial_+ \phi^a\right) - \frac{N}{12\pi}\int_{N_3} \text{Tr}\left[(g^\dagger dg)^3\right] \ . \tag{5.31}$$

The system has a gauge symmetry

$$g_{Ia} \to g_{Ia} e^{-i\lambda^a(x_+,x_-)}, \quad \phi^a \to \phi^a + \lambda^a(x_+, x_-) + \xi^a(x_-) \ . \tag{5.32}$$

We can fix the gauge symmetry by demanding $\phi^a = 0$. This constrains the gauge parameter $\lambda^a(x_+, x_-)$ to be a function that depends only on $x_-$, i.e., $\lambda^a(x_-) = -\xi^a(x_-)$. After imposing the gauge fixing condition, the action (5.31) can be rewritten as a gauged $U(N_f)$ Wess-Zumino-Witten model

$$S = -\frac{N}{4\pi}\int d^2x \,\text{Tr}(\partial_- g^\dagger \partial_+ g) - \frac{N}{12\pi}\int_{N_3} \text{Tr}\left[(g^\dagger dg)^3\right] \ , \tag{5.33}$$

with a $\prod_{a=1}^{N_f} U(1)$ gauge symmetry, that depends only on $x_-$, acting from the right

$$g_{Ia} \to g_{Ia} e^{-i\lambda^a(x_-)} . \tag{5.34}$$

Such chirally gauged Wess-Zumino-Witten models have been studied in [16].

It was shown that the theory has a chiral algebra that consists of a $\mathfrak{u}(N_f)_N$ left-moving chiral algebra and a $\mathfrak{u}(N_f)_N / \prod_{a=1}^{N_f} \mathfrak{u}(1)_N$ right-moving coset chiral algebra. We do not pay attention to the global form of the chiral algebra. The $U(N_f)$ global symmetry of the ultraviolet theory acts on the gauged Wess-Zumino-Witten model from the left $g \to hg$. We can couple the symmetry to the $U(N_f)$ background $B + A\mathbb{1}$. The chiral algebra implies that this symmetry has an 't Hooft anomaly which exactly matches with the anomaly inflow (3.4) from infinity. The theory also has the correct chiral central charge $c_L - c_R = N$.

Let us examine the spectrum of the junction theory. We will first discuss the vertex operators of the left-moving and the right moving chiral algebra and then discuss their pairing. The vertex operators of the $\mathfrak{u}(N_f)$ left-moving chiral algebra are labeled by their $U(N_f)$ representations $\mathcal{R}$. Following the GKO construction [45], the vertex operators of the $\mathfrak{u}(N_f)_N / \prod_{a=1}^{N_f} \mathfrak{u}(1)_N$ right-moving coset chiral algebra are labeled by $U(N_f)$ representations $\mathcal{R}$ and $\prod_{a=1}^{N_f} U(1)$ representations $\{q_a\}$ that the $U(N_f)$ representations $\mathcal{R}$ can be decomposed into. Note that $U(1)$ representations are labeled by integer charges $q$. The $U(N_f)$ Wess-Zumino-Witten model has a diagonal pairing. Following this property, the primary operators of the junction theory pair up vertex operators with the same $U(N_f)$ representations. This suggests that the primary operators are in one-to-one correspondence with the right-moving vertex operators. The labels for the primary operators $(\mathcal{R}, \{q_a\})$ have actual physical meanings. The operator labeled by $(\mathcal{R}, \{q_a\})$ transforms in the $\mathcal{R}$ representations under the $U(N_f)$ global symmetry acting from the left. It is connected to a charge $q_a$ Wilson line of the $U(1)_N$ Chern-Simons theory on the $a$-th interface. It has spin $L_0 - \overline{L}_0 = \sum_a q_a^2 / 2N$.

Let us consider the operator labeled by the $Sym^N(\square)$ representation and $q_1 = N$, $q_2 = \cdots = q_{N_f} = 0$. The operator is connected to transparent Wilson lines of the Chern-Simons theories so it is a genuine local operator. It has one unit of baryon charge and spin $N/2$. The excitations corresponding to this operator can be interpreted as a baryon localized on the junction. Interestingly, the spin of this operator coincides with the spin of the baryon in the same isospin representation.

A similar observation has been employed in the quantum Hall droplet proposal for baryons [15]. It was proposed that on the boundary of the droplet there is a $\mathfrak{u}(N_f)_L$ chiral algebra. The spin of the primary operator in the $Sym^N(\square)$ representation of the $U(N_f)$ global symmetry is also $N/2$, which agrees with the spin of the baryons in the same isospin representation.

## 5.3 $N_f = 2$

We now turn to study the $N_f = 2$ case. As reviewed in section 2.1, the chiral Lagrangian has no ordinary Wess-Zumino term, instead it has a $\mathbb{Z}_2$-valued $\theta$-term with coefficient $N$ mod 2. The Wess-Zumino term of the bulk chiral Lagrangian is crucial in the $N_f > 2$ proposal. It gives rise to $S_{\text{WZW}}$ on the junction, which absorbs the anomaly inflow associated to the $SU(N_f)$ background. Since the Wess-Zumino term is absent in the $N_f = 2$ chiral Lagrangian, one might expect that the $N_f > 2$ proposal does not apply to $N_f = 2$. However, as we will see, the $\theta$-term of bulk chiral Lagrangian plays a similar role as the Wess-Zumino term so the $N_f > 2$ proposal works even at $N_f = 2$.

Consider the case of $\varepsilon \ll 1$. The chiral Lagrangian has a space-depedenet potential for the $SU(2)$ matrix $U$

$$V(U) = -\varepsilon f_\pi^2 \Lambda^3 r \cos(\alpha) \text{Tr}(U) , \tag{5.35}$$

where we use the property $\text{Tr}(U) = \text{Tr}(U^\dagger)$ for $SU(2)$ matrices. The potential depends only on the $x = r \cos(\alpha)$ coordinate so the theory has an emergent translation symmetry along the $y$ direction. To minimize the potential energy, $U = \mathbb{1}$ at large positive $x$ and $U = -\mathbb{1}$ at large negative $x$. The interpolation between them breaks the $SU(2)$ global symmetry, and hence leads a 2+1 dimensional interface perpendicular to the $x$ direction. As discussed in section 2.2, the interface supports a $\mathbb{CP}^1$ sigma model with a $\mathbb{Z}_2$-valued $\theta$-term with coefficient $N$ mod 2. Unlike the case of $N_f > 2$, there is no interface junction at the origin. In particular, the target space at the origin is the same as the target space elsewhere on the interface. It is a happy coincidence that the flag manifold $U(2)/U(1)^2$ is the same as $\mathbb{CP}^1$. As far as the target space is concerned, it is consistent with the $N_f = 2$ application of the $N_f > 2$ proposal.

At certain radius $R_1 \sim 1/(\varepsilon \Lambda)$, $|m| \sim \Lambda$, the chiral Lagrangian is no longer valid so the emergent translation symmetry along the $y$ direction is also not valid. The $\mathbb{CP}^1$ interface then connects with the two interfaces coming from infinity. Together they form a continuous interface along the $y$ axis. Let us pick a continuous orientation along the interface by flipping the orientation as well as the sign of the Chern-Simons level at $y \gg \Lambda$. The interface is described by different theories at different domains. It is described by an $SU(N)_{-1}$ Chern-Simons theory at $y \ll -\Lambda$, an $SU(N)_1$ Chern-Simons theory at $y \gg \Lambda$ and a $\mathbb{CP}^1$ sigma model in between. Let us also keep track of the classical counterterms. We can make the winding counterterm (5.1) jumps at $\alpha = \pi/2, 3\pi/2$:

$$2\pi N \Theta \left( \alpha - \frac{\pi}{2} \right) \left( \frac{F_A \wedge F_A}{8\pi^2} + 2 \frac{\text{Tr}(R \wedge R)}{384\pi^2} \right) +$$
$$2\pi N \Theta \left( \alpha - \frac{3\pi}{2} \right) \left( \frac{F_A \wedge F_A}{8\pi^2} + \frac{\text{Tr}(F_B \wedge F_B)}{8\pi^2} + 2 \frac{\text{Tr}(R \wedge R)}{384\pi^2} \right) . \tag{5.36}$$

This induces a classical Chern-Simons term on the interface. For $y \gg \Lambda$, it is

$$\frac{N}{4\pi} AdA + 2N \mathrm{CS}_{\mathrm{grav}} \; , \tag{5.37}$$

and for $y \ll \Lambda$, it is

$$-\frac{N}{4\pi} AdA - \frac{N}{4\pi} \mathrm{Tr} \left( BdB - \frac{2i}{3} B^3 \right) - 2N \mathrm{CS}_{\mathrm{grav}} \; . \tag{5.38}$$

The $SU(N)_{\pm 1}$ Chern-Simons theory at large $|y|$ can be dualized to a $U(1)_{\mp N}$ Chern-Simons theory. This removes all of the classical Chern-Simons terms except for the one associated to the $SU(2)$ background gauge field $B$ at $y \ll \Lambda$.

We will assume that the transitions between the $\mathbb{CP}^1$ sigma model and the $U(1)_{\pm N}$ Chern-Simons theories are described by a $U(1)_{\pm N}$ Chern-Simons theory coupled to $N_f$ scalars with a spatially varying mass squared. According to the conjectures in [14], the two transitions can both be described by a three-dimensional $SU(N)$ QCD with two quarks at a zero Chern-Simons level. Appendix A.2 studies the interfaces in $SU(N)_{-k+N_f/2}$ Chern-Simon theory coupled to $N_f$ fermions with a spatially varying mass when $N_f > k > 0$. Our discussion here specializes to $N_f = 2$ and $k = 1$.

We can parametrize the $\mathbb{CP}^1$ sigma model by a $U(2)$ matrix $g$ with a block-diagonal $\prod_{a=1}^{2} U(1)$ gauge symmetry acting from the right. Let us introduce an artificial interface in the middle of the Grassmannian sigma model at $y = 0$. We emphasize there are no localized degrees of freedom on the interface. The $\theta$-term of the $\mathbb{CP}^1$ sigma model can be presented as

$$\underbrace{-\frac{N}{12\pi} \int_{N_3} \mathrm{Tr} \left[ (g^\dagger dg)^3 \right]}_{S_{\mathrm{WZW}}} + \frac{N}{4\pi} \int_{y<0} b^1 db^1 - \frac{N}{4\pi} \int_{y>0} b^2 db^2 \; , \tag{5.39}$$

where $b^a = i(g^\dagger dg)_{aa}$ and $N_3$ is a three-dimensional manifold that extends the $x = 0$ locus. The $\theta$ term is independent of the extension $N_3$. It is also independent of the choice of the artificial interface because of the relation

$$\frac{1}{4\pi} b^1 db^1 + \frac{1}{4\pi} b^2 db^2 = -\frac{1}{12\pi} \mathrm{Tr} \left[ (g^\dagger dg)^3 \right] \; . \tag{5.40}$$

The $\theta$ term has many other presentations. For example, it can be presented as

$$\frac{k}{4\pi} \int b^1 db^1, \quad \text{or} \quad -\frac{k}{4\pi} \int b^2 db^2 \; , \tag{5.41}$$

with $k = N \bmod 2$. All these presentations are equivalent on closed manifolds but inequivalent on open manifolds. In particular, they have different gauge variations under the

$\prod_{a=1}^{2} U(1)$ gauge symmetry on open manifolds. At the transition interfaces, we identify $b_-^1$ and $b_-^2$ with the corresponding light cone components of the $U(1)_N$ and the $U(1)_{-N}$ gauge fields, respectively. Among these three presentations, only the gauge variation of (5.39) cancels the gauge variations of the Chern-Simons theories at the transition interfaces. The boundary condition therefore selects the specific presentation of the $\theta$ term in (5.39).

The presentation (5.39) of the $\theta$ term is identical to (5.22). Hence we conclude that the $N_f > 2$ proposal is also valid at $N_f = 2$.

# 6    General Winding Mass Profile

More generally, we can consider a mass profile $m = \varepsilon r e^{if(\alpha,r)} \Lambda^2$ where $f(\alpha,r)$ satisfies $f(\alpha + 2\pi, r) = f(\alpha, r) + 2\pi$. The large radius analysis in section 3 still goes through except now the $N_f$ interfaces are centered at the trajectories where $f(\alpha, r) = (\pi + 2\pi\mathbb{Z})/N_f$. Each of these interfaces still support an $SU(N)_{-1}$ Chern-Simons theory. We are interested in the dynamics in the interior of the space.

For $N_f = 1$, the same discussion in section 4 holds except that the trajectory of the interface is now deformed. For $N_f = 2$, the two interfaces from infinity no longer connect into one interface along the $y$ axis instead they are connected at the origin by a junction which supports a $1 + 1$ dimensional $\mathbb{CP}^1$ sigma model. For $N_f > 2$, there is a possibility that two interfaces can merge into one interface before joining with other interfaces at the origin. This happens when the separation of the two interfaces is comparable to $1/\Lambda$ when they are outside $R_1$ or comparable to $1/\sqrt{m\Lambda}$ when they are inside $R_1$. The merging of the interfaces leads to a new interface junction.

Consider for instance a configuration in figure 6. For simplicity, let us assume that the merging occurs outside $R_1$. The theory on the new interface is an $SU(N)_{-2}$ Chern-Simons theory as discussed in section 2.2. The natural boundary condition on the junction is to identify the $SU(N)$ gauge fields of the three Chern-Simons theories on the interfaces since they originate from the same bulk $SU(N)$ gauge fields. This boundary condition leads to a coset chiral algebra $T_{1,1}^N$ on the junction [38]. Here we define the chiral algebra

$$T_{k_1,\cdots,k_n}^N = \left[ \frac{\mathfrak{su}(N)_{k_1} \times \cdots \times \mathfrak{su}(N)_{k_n}}{\mathfrak{su}(N)_{k_1+\cdots+k_n}} \right]_R . \tag{6.1}$$

The new interface continues towards the origin. We can dualize the $SU(N)_{-2}$ Chern-Simons theory to a $U(2)_N$ Chern-Simons theory by adding twice of the classical counterterm (4.3).

Let us first consider the situation when $\varepsilon \ll 1$. At radius $R_1 \sim 1/(\varepsilon\Lambda)$, $|m| \sim \Lambda$, all the interfaces undergo a transition to a $\mathbb{CP}^{N_f - 1}$ sigma model except for the new interface,

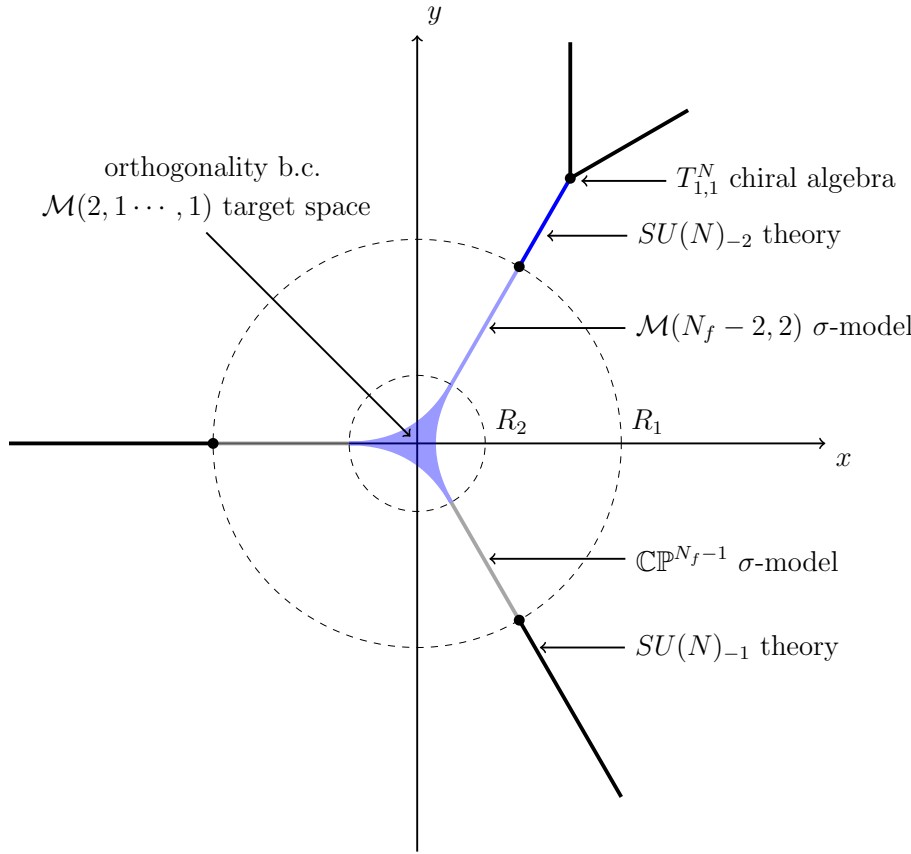

Figure 6: The dynamics in the interior for a mass profile $m = \varepsilon r e^{if(\alpha,r)} \Lambda^2$ with $\varepsilon \ll 1$ when $N_f = 4$. The centers of the interfaces follow the trajectories (colored in black and blue) where $f(\alpha, r) = (\pi + 2\pi\mathbb{Z})/N_f$. Two interfaces (colored in black) can merge into one interface (colored in blue) when their separation is less than $1/\Lambda$ outside the radius $R_1 \sim 1/(\varepsilon\Lambda)$. This leads to an interface junction that supports the chiral algebra $T_{1,1}^N$ defined in (6.1). The theories on the interfaces undergo a transition from Chern-Simons theories to sigma models with appropriate Wess-Zumino terms at radius $R_1$. It leads to chiral modes localized at the transition interfaces. The interfaces form a junction of size $R_2 \sim 1/(\varepsilon^{1/3}\Lambda)$ at the origin. On the interface junction, one should impose an orthogonality boundary condition that reduces the target space on the junction to the flag manifold $\mathcal{M}(2, 1, \cdots, 1)$. The flag sigma model on the junction is supplemented by a Wess-Zumino term $S_{\mathrm{WZW}}$ defined in (5.22).

which undergoes a transition to a Grassmannian sigma model with target space

$$\mathcal{M}(N_f - 2, 2) = \frac{U(N_f)}{U(2) \times U(N_f - 2)} \ . \tag{6.2}$$

Here we introduce a notation for general flag manifolds

$$\mathcal{M}(n_1, \cdots, n_k) = \frac{U(n_1 + \cdots + n_k)}{U(n_1) \times \cdots \times U(n_k)} \ . \tag{6.3}$$

The $\mathcal{M}(N_f - 2, 2)$ sigma model has a three-dimensional Wess-Zumino term with coefficient $N$ (see appendix A.2). We will assume that the transition on the new interface is described by a $U(2)_N$ Chern-Simons theory coupled to $N_f$ scalars. As discussed in appendix A.2, this leads to a chiral boundary condition. After integrating out the $U(2)$ Chern-Simons gauge field and fixing the gauge symmetry, the $U(2)$ gauge symmetry of the $\mathcal{M}(N_f - 2, 2)$ sigma model becomes restricted on the boundary such that its gauge parameter depends only on one of the light cone coordinates. This leads to chiral modes at the transition interface.

Eventually all the interfaces meet at the origin and form an interface junction with size $R_2$. Similar to the discussion in section 5, on the junction, we should impose an orthogonality boundary condition, which reduces the target space to

$$\mathcal{M}(2, 1, \cdots, 1) = \frac{U(N_f)}{U(2) \times \prod_{a=1}^{N_f - 2} U(1)} \ . \tag{6.4}$$

The $\mathcal{M}(2, 1, \cdots, 1)$ sigma model can be parametrized by a $U(N_f)$ matrix with a block-diagonal $U(2) \times \prod_{a=1}^{N_f - 2} U(1)$ gauge symmetry acting from the right. It is supplemented by a Wess-Zumino term $S_{\mathrm{WZW}}$ of the $U(N_f)$ matrix defined in (5.22). The Wess-Zumino term $S_{\mathrm{WZW}}$ is not gauge invariant. Its gauge non-invariance is canceled by the three-dimensional Wess-Zumino terms of the sigma models on the intefaces. Similar to the discussion in section 5, the Wess-Zumino term $S_{\mathrm{WZW}}$ on the junction can be derived from the Wess-Zumino term $N\Gamma_{\mathrm{WZ}}$ of the bulk chiral Lagrangian.

When $\varepsilon \gtrsim 1$, the Chern-Simons theories on the interfaces directly couple to the 1+1 dimensional $\mathcal{M}(2, 1, \cdots, 1)$ sigma model at the center. After integrating out the Chern-Simons gauge fields and fixing the gauge symmetries, the theory on the junction is reduced to a gauged $U(N_f)$ Wess-Zumino-Witten model with a restricted block-diagonal $U(2) \times \prod_{a=1}^{N_f - 2} U(1)$ gauge symmetry acting from the right. The gauge parameters depend only on one of the light coordinates. The theory has a $S_{2,1,\cdots,1}^N$ chiral algebra. Here we define a chiral algebra $S_{n_1,\cdots,n_k}^N$ which consists of a $\mathfrak{u}(\sum n_a)_N$ left-moving chiral algebra and a $\mathfrak{u}(\sum n_a)_N / \prod \mathfrak{u}(n_a)_N$ right-moving chiral algebra. The $U(N_f)$ global symmetry of the ultraviolet theory acts on the gauged Wess-Zumino-Witten model from the left. Hence the chiral algebra implies an 't Hooft anomaly of the global symmetry which is consistent with

the anomaly inflow (3.4). The total chiral central charge of the two junction theories is $c_L - c_R = N$ which also matches with the anomaly inflow from infinity.

It is straightforward to generalize the discussions to general mass profile $m = \varepsilon r e^{if(\alpha,r)}\Lambda^2$. The interfaces can form a network connected by interface junctions. In particular there is always an interface junction at the origin. We will consider only $\varepsilon \gtrsim 1$ for simplicity. The theories on the interfaces then are all Chern-Simons theories. The theory on the junction at the origin is a chirally gauged $U(N_f)$ Wess-Zumino-Witten model whose gauge symmetry is determined by the Chern-Simons theories on the interfaces that the junction connects to. The gauge parameters are restricted such that they depend only on one of the light cone coordinates. The chirally gauged Wess-Zumino-Witten model has a chiral algebra in the form of $S^N_{n_1,\cdots,n_k}$, which has a $U(N_f)$ anomaly that absorbs the anomaly inflow (3.4). The theories on the junctions away from the origin have chiral algebras in the form of $T^N_{k_1,\cdots,k_n}$. The total chiral central charge of these junction theories is always $c_L - c_R = N$ as constrained by the anomaly inflow.

# Acknowledgements

We thank Clay Crdova for collaboration in the early stages of this project and many stimulating discussions. We especially thank Nathan Seiberg for advice, encouragement, numerous helpful discussions throughout the project and comments on the draft. We thank Po-Shen Hsin for useful conversations. We thank Yunqin Zheng for comments on the draft. P.G. is supported by Physics Department of Princeton University. H.T.L. is supported by a Croucher Scholarship for Doctoral Study, a Centennial Fellowship from Princeton University and Physics Department of Princeton University.

# A  Interface in Chern-Simons-Matter Theory

In this appendix, we study interfaces in Chern-Simons-Matter theories including $U(1)_k$ Chern-Simons theory coupled to $N_f$ scalars and $SU(N)_{-k+N_f/2}$ Chern-Simons theory coupled to $N_f$ fermions with $N_f > k > 0$.

## A.1  $U(1)_k + N_f$ Scalars

We will consider interfaces in $U(1)_k$ Chern-Simons theory coupled to $N_f$ complex scalars with an $SU(N_f)$-symmetric potential. We make the mass squared of the scalars monotonically decreases in one spatial coordinate $x$, from a positive value to a negative value. As a result, the scalars develop a non-zero position-dependent vacuum expectation value

$v(x)$ everywhere. $v(x)$ smoothly increases from zero to a finite value $v_0$. Ignoring the massive amplitude fluctuation of the scalars, the low energy dynamics of the system can be described by an $\mathbb{S}^{2N_f-1}$ sigma model coupled to a dynamical gauge field $a$ with Lagrangian

$$\mathcal{L} = \frac{1}{4g^2}F^{\mu\nu}F_{\mu\nu} + \frac{k}{4\pi}\epsilon^{\mu\nu\rho}a_\mu\partial_\nu a_\rho + v(x)^2\left(\partial^\mu\Phi_I^\dagger\partial_\mu\Phi_I - 2ia_\mu\Phi_I^\dagger\partial^\mu\Phi_I + a_\mu a^\mu\right) + v'(x)^2 \quad (A.1)$$

where $\Phi_I$ are complex scalars that satisfy the constraint $\sum|\Phi_I|^2 = 1$. The $U(1)$ gauge symmetry acts as $\Phi_I \to \Phi_I e^{-i\lambda}$, $a \to a + d\lambda$. The transverse coordinates are denoted by $y$ and $t$, and we pick the convention $\epsilon^{xyt} = -1$.

### A.1.1 Localized Chiral Mode

Let us start by analyzing the equation of motion of the gauge field:

$$\partial_\nu F^{\mu\nu} + m_{CS}\epsilon^{\mu\nu\rho}\partial_\nu a_\rho + m_H^2(x)a^\mu = m_H(x)^2 b^\mu \ , \quad (A.2)$$

where we define $m_{CS} = \frac{1}{2\pi}g^2 k$, $m_H(x) = \sqrt{2}gv(x)$ and $b = \sum i\Phi_I^\dagger d\Phi_I$. In the light cone coordinate,

$$x_\pm = \frac{1}{2}(t \pm y), \quad \partial_\pm = \partial_t \pm \partial_y, \quad a_\pm = a_t \pm a_y \ , \quad \epsilon^{x-+} = 1, \quad (A.3)$$

the equation of motion becomes

$$\partial_+\partial_- a_x - \frac{1}{2}\partial_x(\partial_- a_+ + \partial_+ a_-) + \frac{1}{2}m_{CS}(\partial_- a_+ - \partial_+ a_-) + m_H^2(x)a_x = m_H(x)^2 b_x \ ,$$

$$\frac{1}{2}(\partial_+\partial_- - 2\partial_x^2)a_\pm - \frac{1}{2}\partial_\pm^2 a_\mp + \partial_\pm\partial_x a_x \pm m_{CS}(\partial_x a_\pm - \partial_\pm a_x) + m_H^2(x)a_\pm = m_H(x)^2 b_\pm \ .$$

$$(A.4)$$

These equations are invariant under a parity transformation

$$y \to -y, \quad a_\pm \to a_\mp, \quad b_\pm \to b_\mp, \quad m_{CS} \to -m_{CS} \ . \quad (A.5)$$

The solutions of the equations of motion can be decomposed into a nonhomogeneous solution and homogeneous solutions with arbitrary coefficients. The homogeneous solutions solve the equations with $b = 0$. They represent dynamical excitations of the gauge field. We are interested in gapless excitations localized on the interface. They can be decomposed into chiral and anti-chiral modes. Without lost of generality, we will assume positive chirality

$\partial_- a_\mu = 0$. Correspondingly, the homogeneous equation simplifies to

$$\partial_x^2 a_- + m_{CS}\partial_x a_- - m_H^2(x)a_- = 0 ,$$
$$\partial_x\partial_+ a_- + m_{CS}\partial_+ a_- - 2m_H^2(x)a_x = 0 , \qquad \text{(A.6)}$$
$$\partial_x^2 a_+ + \frac{1}{2}\partial_+^2 a_- - \partial_+\partial_x a_x - m_{CS}(\partial_x a_+ - \partial_+ a_x) - m_H^2(x)a_+ = 0 .$$

The equations have nontrivial bounded solutions only for positive $m_{CS}$. The solutions are in the form of

$$a_x = a_- = 0, \quad a_+ = a_+(x_+, x) . \qquad \text{(A.7)}$$

Similarly, using parity transformation (A.5), the anti-chiral solutions exist only for negative $m_{CS}$.

To find the bounded solution, we first assume $a_x = a_- = 0$, which reduces (A.6) to

$$\partial_x^2 a_+ - m_{CS}\partial_x a_+ - m_H^2(x)a_+ = 0 . \qquad \text{(A.8)}$$

The equation is independent of the transverse coordinates so $a_+$ can be decomposed into $a_+ = \partial_+\phi(x_+)h(x)$. Let us examine the behavior of $h(x)$ in the asymptotic regions:

- When $x \to +\infty$, $m_H^2(x) \to m_H^2(+\infty)$, the equation implies the following asymptotic behavior

$$h''(x) - m_{CS}h'(x) - m_H^2(+\infty)h(x) = 0 \quad \Rightarrow \quad h(x) = c_1 e^{\lambda_+ x} + c_2 e^{\lambda_- x} , \qquad \text{(A.9)}$$

  where $\lambda_\pm = \frac{1}{2}(m_{CS} \pm \sqrt{m_{CS}^2 + 4m_H^2(+\infty)})$. We set $c_1 = 0$ for the solutions to be bounded. This selects a unique solution.

- When $x \to -\infty$, $m_H^2(x) \to 0$, the equation implies the following asymptotic behavior

$$h''(x) - m_{CS}h'(x) = 0 \quad \Rightarrow \quad h(x) = c_3 e^{m_{CS}x} + c_4 . \qquad \text{(A.10)}$$

  For bounded solutions with $c_1 = 0$, $c_3$ and $c_4$ generally do not vanish. Hence the equation has a bounded solution only if $m_{CS}$ is positive.

The assumption $a_x = a_- = 0$ can be justified by considering the first two equations of (A.6). The first equation of (A.6) is the same as (A.8) with $m_{CS} \to -m_{CS}$ so the equation can have nontrivial bounded solutions only if $m_{CS}$ is negative. However, for such solutions, $(\partial_x + m_{CS})a_-$ approaches a constant when $x \to -\infty$ which makes $a_x$ diverge according to the second equation of (A.6). Therefore bounded solutions have $a_x = a_- = 0$.

We conclude that when $\pm m_{CS}$ is positive, the equations of motion have a homogeneous

solution

$$a_x = a_\mp = 0, \quad a_\pm = \partial_\pm \phi(x_\pm)h(x) \ , \tag{A.11}$$

which represents a compact chiral boson localized in the $x$ coordinate. Note that it is not a free compact boson when $N_f > 1$. It interacts with other gapless fields.

### A.1.2  Sharp Interface

Let us consider the limit $g$ approaching infinity and $v(x)$ approaching a step function $v_0\Theta(x)$ with large $v_0$. The system reduces to a sharp interface between a $U(1)_k$ Chern-Simons theory

$$\frac{k}{4\pi}\int_{x<0} d^3x\, \epsilon^{\mu\nu\rho}a_\mu\partial_\nu a_\rho = \frac{k}{2\pi}\int_{x<0} d^3x\left(a_x(\partial_-a_+ - \partial_+a_-) + \frac{1}{2}(a_+\partial_x a_- - a_-\partial_x a_+)\right), \tag{A.12}$$

and a $\mathbb{CP}^{N_f-1}$ sigma model with a Wess-Zumino term

$$\int_{x>0} d^3x\left(v_0^2 D^\mu\Phi_I^\dagger D_\mu\Phi_I + \frac{k}{4\pi}\epsilon^{\mu\nu\rho}b_\mu\partial_\nu b_\rho\right) \ , \tag{A.13}$$

where $D\Phi_I = (\partial + ib)\Phi_I$ and $b = \sum i\Phi_I^\dagger d\Phi_I$. When $N_f = 1$, the $\mathbb{CP}^{N_f-1}$ sigma model is simply a trivial theory.

In general, one can impose many different boundary conditions on the sharp interface but here we will land on a unique boundary condition by taking the limit described above.

The limit has two consequences on the equations of motion (A.2)

- The profile $h(x)$ becomes a step function $\Theta(x)$. When $\pm m_{CS}$ is positive, the homogeneous solution has $a_\mp = 0$ and $a_\pm = \partial_\pm\phi(x_\pm)\Theta(x)$. This means that $a_\mp$ is always continuous across the interface while $a_\pm$ can have a discontinuity.

- The equation of motion (A.2) becomes $a_\mu = b_\mu$ for $x < 0$.

When $\pm k$ is positive, the appropriate boundary condition on the interface is $a_\mp = b_\mp$. There are no constraints relating $a_\pm$ and $b_\pm$ since they can differ by a discontinuous homogeneous solution with arbitrary coefficient.

We now discuss the consequences of this boundary condition. In the discussion below, we will assume $k$ is positive. The case with negative $k$ can be inferred using the parity transformation (A.5). The boundary condition $a_- = b_-$ can be imposed dynamically by the equations of motion. This amounts to adding a term on the interface

$$\frac{k}{2\pi}\int_{x=0} d^2x\left(a_+b_- - \frac{1}{2}(a_-a_+ + b_-b_+)\right) \ . \tag{A.14}$$

The full action, including (A.13), (A.12) and (A.14), is invariant under the gauge symmetry

$$\Phi_I \to \Phi_I e^{-i\lambda}, \quad a \to a + d\lambda, \quad b \to b + d\lambda . \tag{A.15}$$

To check the gauge invariance, let us compute the gauge variation of a piece of the action

$$S = -\frac{k}{4\pi} \int_{x=0} d^2x \, a_- a_+ \pm \frac{k}{4\pi} \int_{x<0} d^3x \, \epsilon^{\mu\nu\rho} a_\mu \partial_\nu a_\rho. \tag{A.16}$$

Here $\pm$ sign is introduced so that the computation can also apply to the term associated to $b$. The gauge variation can be written as a boundary term

$$S \to S - \frac{k}{2\pi} \int_{x=0} d^2x \left( a_\pm \partial_\mp \lambda + \frac{1}{2} \partial_- \lambda \partial_+ \lambda \right) . \tag{A.17}$$

It is then straightforward to the check the gauge invariance of the full action. Let us also compute the variation of (A.16) with respect to $a$, which gives

$$\delta S = \pm \frac{k}{2\pi} \int_{x<0} d^3x \, (f_{-+} \delta a_x + f_{x-} \delta a_+ + f_{+x} \delta a_-) - \frac{k}{2\pi} \int_{x=0} d^2x \, a_\mp \delta a_\pm . \tag{A.18}$$

Combing it with the variation of $\frac{k}{2\pi} \int d^2x \, a_+ b_-$ gives the constraint $a_- = b_-$.

We can integrate out $a_-$. It restricts the remaining gauge fields to be pure gauge

$$a_x = \partial_x \phi, \quad a_+ = \partial_+ \phi , \tag{A.19}$$

where $\phi$ is a compact boson, $\phi \sim \phi + 2\pi$, with a redundancy $\phi \to \phi + \xi(x_-)$. Substituting the solutions back to (A.13), (A.12) and (A.14) gives

$$\int_{x>0} d^3x \left( v_0^2 D^\mu \Phi_I^\dagger D_\mu \Phi_I + \frac{k}{4\pi} \epsilon^{\mu\nu\rho} b_\mu \partial_\nu b_\rho \right) - \frac{k}{4\pi} \int_{x=0} d^2x \, (\partial_- \phi \partial_+ \phi + b_- b_+ - 2b_- \partial_+ \phi) . \tag{A.20}$$

Even though the boundary action is not chiral, $\phi$ is in fact a chiral boson due to the redundancy $\phi \to \phi + \xi(x_-)$. Let us summarize the gauge symmetry of the system

$$\Phi_I \to \Phi_I e^{-i\lambda(x_-,x_+)}, \quad b \to b + d\lambda(x_-,x_+), \quad \phi \to \phi + \lambda(x_-,x_+) + \xi(x_-) . \tag{A.21}$$

We can fix the gauge symmetry by demanding $\phi = 0$. It reduces the action to

$$\int_{x>0} d^3x \left( v_0^2 D^\mu \Phi_I^\dagger D_\mu \Phi_I + \frac{k}{4\pi} \epsilon^{\mu\nu\rho} b_\mu \partial_\nu b_\rho \right) - \frac{k}{4\pi} \int_{x=0} d^2x \, b_- b_+ . \tag{A.22}$$

The gauge parameter $\lambda$ is restricted such that it depends only on $x_-$, $\lambda(x_-) = -\xi(x_-)$.

The restricted gauge symmetry on the boundary can be interpreted as an extension of the $\mathbb{CP}^{N_f-1}$ sigma model to the $\mathbb{S}^{2N_f-1}$ sigma model by the compact chiral boson (A.11) [46]. When $N_f = 1$, the bulk sigma model is trivial so the theory on the interface is a free compact chiral boson with the action

$$-\frac{k}{4\pi} \int_{x=0} d^2x\, \partial_-\phi\partial_+\phi \;, \tag{A.23}$$

and the gauge symmetry $\phi \to \phi + \lambda(x_-)$.

We can couple the system to a $U(1)$ background gauge field $A$ by introducing the following coupling to the full Lagrangian (A.1)

$$\frac{1}{2\pi}\epsilon^{\mu\nu\rho}A_\mu\partial_\nu a_\rho \;. \tag{A.24}$$

When the system is reduced to a sharp interface, the coupling becomes

$$\frac{1}{2\pi} \int_{x<0} d^3x\, \epsilon^{\mu\nu\rho}A_\mu\partial_\nu a_\rho + \frac{1}{2\pi} \int_{x>0} d^3x\, \epsilon^{\mu\nu\rho}A_\mu\partial_\nu b_\rho + \frac{1}{2\pi} \int_{x=0} d^2x\, \epsilon^{\mu\nu}A_\mu(b_\nu - a_\nu) \;. \tag{A.25}$$

The background gauge field $A$ couples to the $U(1)$ magnetic symmetry of the Chern-Simons theory and the baryon current of the sigma model. As discussed above, when $k$ is positive, we can integrate out $a_-$ and solve the constraint by introducing a compact boson $\phi$

$$a_x = -\frac{1}{k}A_x + \partial_x\phi, \quad a_+ = -\frac{1}{k}A_+ + \partial_+\phi \;. \tag{A.26}$$

This generates the following coupling in addition to the action (A.20)

$$\frac{1}{2\pi} \int_{x>0} d^3x\, \epsilon^{\mu\nu\rho}A_\mu\partial_\nu b_\rho - \frac{1}{2\pi} \int_{x=0} d^2x\, A_-(b_+ - \partial_+\phi) \;. \tag{A.27}$$

As before, we can fix the gauge by setting $\phi = 0$. This reduces the coupling to

$$\frac{1}{2\pi} \int_{x>0} d^3x\, \epsilon^{\mu\nu\rho}A_\mu\partial_\nu b_\rho - \frac{1}{2\pi} \int_{x=0} d^2x\, A_-b_+ \;. \tag{A.28}$$

When $N_f = 1$, the bulk sigma model is trivial so the background gauge field $A$ only couples to the compact boson on the interface

$$\frac{1}{2\pi} \int_{x=0} d^2x\, A_-\partial_+\phi \;. \tag{A.29}$$

$A$ can be interpreted as the background gauge field of the shift symmetry of the compact chiral boson.

## A.2  $SU(N)_{-k+N_f/2} + N_f$ Fermions

Consider an $SU(N)_{-k+N_f/2}$ Chern-Simons theory coupled to $N_f$ degenerate fermions with $N_f > k > 0$. According to [14], the theory flows to an $SU(N)_{-k+N_f}$ Chern-Simons theory for large positive mass, an $SU(N)_{-k}$ Chern-Simons theory for large negative mass, and a quantum phase, described by a sigma model with a Grassmannian target space

$$\mathcal{M}(k, N_f - k) = \frac{U(N_f)}{U(k) \times U(N_f - k)} \ , \tag{A.30}$$

for intermediate mass. Let us also keep track of the classical counterterms of the background gauge fields. The theory has a $U(N_f)/\mathbb{Z}_N$ global symmetry [47]. For simplicity, we will only couple the theory to a $U(1)$ background gauge field $A$ and an $SU(N_f)$ background gauge field $B$. The two gauge fields can be combined into a $U(N_f)$ gauge field $B + A\mathbb{1}$. The classical counterterms at large positive and large negative mass differ by

$$\frac{NN_f}{4\pi} AdA + \frac{N}{4\pi} \mathrm{Tr}\left(BdB - \frac{2i}{3}B^3\right) + 2NN_f \mathrm{CS}_{\mathrm{grav}} \ . \tag{A.31}$$

We can use level-rank duality [37] to dualize the $SU(N)_{-k+N_f}$ and the $SU(N)_{-k}$ Chern-Simons theory of the two asymptotic phases to $U(N_f - k)_{-N}$ and $U(k)_N$, respectively. This removes the difference of the $U(1)$ and the gravitational counterterms but leaves the $SU(N_f)$ counterterms intact.

The transitions between the quantum phase and the asymptotic phases are described by two separate bosonic theories: a $U(k)_N$ coupled to $N_f$ scalars and a $U(N_f - k)_{-N}$ coupled to $N_f$ scalars. The Grassmannian sigma model of the quantum phase can be seen from the bosonic theories by condensing the scalars. The Chern-Simons terms of the bosonic theories become the Wess-Zumino term of the sigma model (or the $\mathbb{Z}_2$-valued $\theta$ term of the $\mathbb{CP}^1$ sigma model when $N_f = 2, k = 1$).

Let us explain the Wess-Zumino term in more detail. The Grassmannian sigma model can be described by a $U(N_f)$ matrix $g$ with a block-diagonal $U(k) \times U(N_f - k)$ gauge symmetry acting from the right. We define two composite gauge fields

$$\begin{aligned} a_{IJ} &= i(g^\dagger dg)_{IJ}, & 1 \le I, J \le k \ , \\ b_{IJ} &= i(g^\dagger dg)_{IJ}, & k+1 \le I, J \le N_f \ . \end{aligned} \tag{A.32}$$

They transform as ordinary gauge fields under the $U(k)$ and the $U(N_f - k)$ gauge symmetry respectively. The Wess-Zumino term is defined on a four-dimensional manifold $N_4$ which

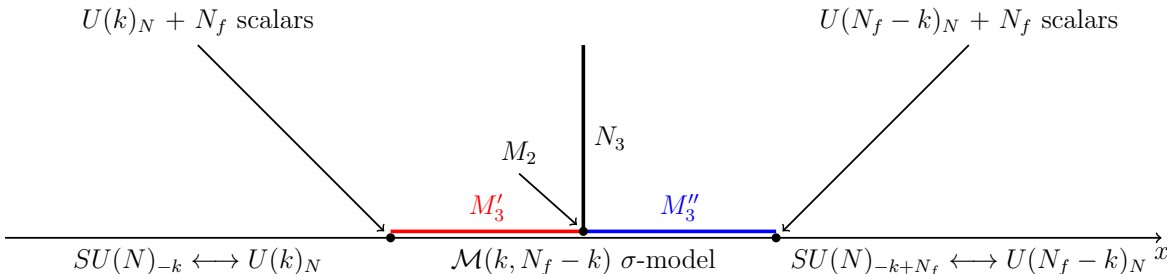

Figure 7: $SU(N)_{-k+N_f/2}$ Chern-Simons theory coupled to $N_f$ fermions with a spatially varying mass $m \propto x$. There are two interfaces separating the $U(k)_N$ Chern-Simons theory, the $\mathcal{M}(k, N_f - k)$ Grassmannian sigma model and the $U(N_f - k)_{-N}$ Chern-Simons theory. The interfaces can be described by the bosonic theories with a spatially varying mass squared. The sigma model has a Wess-Zumino term which consists of three components: $NCS(a)$ on $M_3'$ (red), $NCS(b)$ on $M_3''$ (blue) and $NWZW(g)$ on $N_3$ (black).

extends the original spacetime $M_3$,

$$\frac{N}{4\pi} \int_{N_4} \sum_{\substack{1 \leq I, K \leq k \\ k+1 \leq J, L \leq N_f}} (g^\dagger dg)_{IJ}(g^\dagger dg)_{JK}(g^\dagger dg)_{KL}(g^\dagger dg)_{LI} \ . \tag{A.33}$$

The definition is independent of the extensions modulo $2\pi$. The integrand is a total derivative so the Wess-Zumino term can also be expressed as a Chern-Simons term for the composite gauge fields

$$N \int_{M_3} \mathrm{CS}(a) = -N \int_{M_3} \mathrm{CS}(b) \ . \tag{A.34}$$

$\mathrm{CS}(a)$ denotes the Chern-Simons form of $a$

$$\mathrm{CS}(a) = \frac{1}{4\pi} \mathrm{Tr}\left(ada - \frac{2i}{3}a^3\right) \ . \tag{A.35}$$

When $N_f = 2$, $k = 1$, the $\mathbb{CP}^1$ sigma model does not have a Wess-Zumino term, instead, it has a $\mathbb{Z}_2$-valued discrete $\theta$ term [33, 34] which can also be expressed as (A.34).

Let us now make the fermion mass vary slowly along one coordinate, $m \propto x$. As illustrated in figure 7, this leads to two interfaces that separate the $U(k)_N$ Chern-Simons theory, the Grassmannian sigma model and the $U(N_f - k)_{-N}$ Chern-Simons theory. In the bosonic descriptions, the mass squared of the scalars varies across the interfaces. This leads to chiral boundary conditions on the interfaces. The boundary conditions identify $a_-$ and $b_-$ of the Grassmannian sigma model with the corresponding light cone components of the $U(k)$ and the $U(N_f - k)$ gauge fields on the two interfaces, respectively. Similar to the discussions in appendix A.1, we can integrate out the Chern-Simons gauge fields and

fix the gauge symmetry. This leads to a restricted $U(k)$ and $U(N_f - k)$ gauge symmetry on the two interfaces, respectively. The gauge parameters are restricted such that they can depend only on the light cone coordinate $x_-$ on the interfaces. Hence there are chiral modes localized on the interfaces. These chiral modes do not carry $SU(N_f)$ anomaly.

Let us examine the Grassmannian sigma model in the intermediate region. Around the two interfaces, the Chern-Simons terms of the dual bosonic theories both descend to the Wess-Zumino term of the sigma model. However, they have different presentations. One of them is $N\mathrm{CS}(a)$ and the other one is $-N\mathrm{CS}(b)$. This suggests us to use a specific presentation of the Wess-Zumino term that agrees with both presentations around the interfaces. Let us introduce an artificial interface $M_2$ in the middle of the Grassmannian sigma model. There are no localized degrees of freedom on the interface. As illustrated in figure 7, we define the manifold on the left hand side by $M_3'$, the manifold on the right hand side by $M_3''$ and a three-dimensional manifold that extends $M_2$ by $N_3$. We can present the Wess-Zumino term as

$$N \int_{M_3'} \mathrm{CS}(a) - N \int_{M_3''} \mathrm{CS}(b) + N \int_{N_3} \mathrm{WZW}(g) \ . \tag{A.36}$$

where $\mathrm{WZW}(g)$ denotes

$$\mathrm{WZW}(g) = -\frac{1}{12\pi} \mathrm{Tr}\left[(g^\dagger dg)^3\right] = \mathrm{CS}(a) + \mathrm{CS}(b) \ . \tag{A.37}$$

The Wess-Zumino term in this presentation is independent of the extensions on $N_3$. When $N_f = 2$, $k = 1$, the $\theta$ term of the $\mathbb{CP}^1$ sigma model can also be expressed as (A.36).

The three different presentations of the Wess-Zumino term in (A.34) and (A.36) are equivalent when $M_3 = M_3' \cup M_3''$ is a closed manfiold. It can be shown using the relation (A.37). But these presentations are not equivalent on open manifolds. In particular, they have different gauge variations on open manifolds. We choose the presentation (A.36) such that its gauge variation cancels the gauge variations of the Chern-Simons theories at the interfaces.

We can couple the $SU(N_f)$ global symmetry of the Grassmannian sigma model, $g \to hg$, to the $SU(N_f)$ background gauge field $B$. This modifies the Wess-Zumino term to

$$N \int_{M_3'} \mathrm{CS}(\widetilde{a}) - N \int_{M_3''} \mathrm{CS}(\widetilde{b}) + N \int_{N_3} \mathrm{WZW}(g) - \frac{N}{4\pi} \int_{M_2} \mathrm{Tr}(iBdgg^\dagger) \ , \tag{A.38}$$

where we define

$$\begin{aligned}
\widetilde{a}_{IJ} &= i\left(g^\dagger(d + iB)g\right)_{IJ}, & 1 &\le I, J \le k \ , \\
\widetilde{b}_{IJ} &= i\left(g^\dagger(d + iB)g\right)_{IJ}, & k+1 &\le I, J \le N_f \ .
\end{aligned} \tag{A.39}$$

The modified Wess-Zumino term is not invariant under the infinitesimal gauge symmetry

$$B \to B + [B, \zeta] + d\zeta, \quad g \to g + i\zeta g .$$

(A.40)

It is shifted by

$$\frac{N}{4\pi} \text{Tr}(Bd\zeta) ,$$

(A.41)

which exactly cancels the anomaly inflow due to the difference in the $SU(N_f)$ counterterms (A.31) at large $|x|$.

When the fermion mass varies rapidly, the intermediate region described by the Grassmannian sigma model shrinks and becomes negligible. The two interfaces then collapse into one interface between the $U(k)_N$ and the $U(N_f - k)_{-N}$ Chern-Simons theory. The theory on the interface is a two-dimensional Grassmannian sigma model with the same target space $\mathcal{M}(k, N_f - k)$. The Wess-Zumino term of the three-dimensional sigma model reduces

$$N \int_{N_3} \text{WZW}(g) ,$$

(A.42)

where $N_3$ extends the interface worldvolume $M_2$. On the interface, we impose boundary conditions that identifies $a_-$ and $b_-$ with the corresponding light cone component of the $U(k)$ and the $U(N_f - k)$ gauge fields, respectively. After integrating out the Chern-Simons gauge fields and fixing the gauge symmetry, the theory on the interface becomes a gauged $U(N_f)$ Wess-Zumino-Witten model with a restricted $U(k) \times U(N_f - k)$ gauge symmetry acting from the right. The gauge parameters can depend only on the $x_-$ coordinate.

The theory has a chiral algebara that consists of a $\mathfrak{u}(N_f)_N$ left-moving chiral algebra and a $\mathfrak{u}(N_f)_N / (\mathfrak{u}(k)_N \times \mathfrak{u}(N_f - k)_N)$ right-moving coset chiral algebra. The chiral algebra has an 't Hooft anomaly for the $U(N_f)$ background that couples only to the left-moving currents. The 't Hooft anomaly exactly absorbs the $U(N_f)$ anomaly inflow due to the difference in the classical counterterms (A.31) at large $|x|$. The chiral central charge of the chiral algebra also agrees with the gravitational anomaly inflow which includes contributions from the $SU(N)_{-k}$ and $SU(N)_{-k+N_f}$ Chern-Simons theories as well as the difference in the classical counterterms.

# B   Defects in Real Scalar Theory

In this appendix, we shall prove the claim made at the end of section 4: the 1+1 dimensional effective mass of the $\eta'$ particle [localized on the *string* at $(x, y) = (x_0, 0)$; we shall choose $x_0 = 0$] is positive when $\kappa(y)$ and $\mu^2(x)$ are discontinuous step functions. The Lagrangian

that we are going to study is (4.12)

$$\mathcal{L} = \frac{1}{2}(\partial\Phi)^2 + \kappa(y)\Phi + \frac{1}{2}\mu^2(x)\Phi^2 + \frac{1}{4}\lambda\Phi^4 \ , \tag{B.1}$$

where we define the scalar $\Phi = f_{\eta'}\eta'$ with a canonical mass dimension, and rescale $\kappa(y)$ and $\mu^2(x)$ accordingly. We restrict $\kappa(y)$ to be an odd function of $y$ since the original problem has the same property. Hence, the above Lagrangian has a $\mathbb{Z}_2$ symmetry, $y \to -y$ and $\Phi \to -\Phi$.

## B.1 Interface

To get a sense of the kind of arguments we will use to prove the above claim, we shall first consider an *interface* where $\kappa(y) = 0$, i.e., only $\mu^2(x)$ varies along $x$. One may expect that because $\mu^2(0) = 0$, $\eta'$ becomes a gapless 2+1 dimensional excitation localized along $x = 0$. However, we shall see that when $\mu^2(x)$ is a discontinuous step function, the 2+1 dimensional effective mass of $\eta'$ is always positive.

Let $\mu^2(x)$ be a step function

$$\mu^2(x) = \begin{cases} +\mu_1^2 > 0, & x > 0 \\ -\mu_2^2 < 0, & x < 0 \end{cases} . \tag{B.2}$$

The vacuum solution $v(x)$ satisfies the equation of motion

$$-\partial_x^2 v(x) + \mu^2(x)v(x) + \lambda v(x)^3 = 0 \ , \tag{B.3}$$

where we expect that, for energy reasons, the vacuum depends only on the longitudinal direction $x$. We also expect that $v(x) = \mu_2/\sqrt{\lambda}$ as $x \to -\infty$, and it decays to $v(x) = 0$ as $x \to +\infty$. In fact, the above equation can be exactly solved, and the solution is

$$v(x) = \begin{cases} -\dfrac{\mu_2}{\sqrt{\lambda}}\tanh\left[\dfrac{\mu_2 x}{\sqrt{2}} - \operatorname{arctanh}\left(\dfrac{\sqrt{\lambda}v_0}{\mu_2}\right)\right] \ , & x < 0 \\[4mm] \dfrac{\sqrt{2}\mu_1}{\sqrt{\lambda}}\operatorname{csch}\left[\mu_1 x + \operatorname{arcsinh}\left(\dfrac{\sqrt{2}\mu_1}{v_0\sqrt{\lambda}}\right)\right] \ , & x > 0 \end{cases} \tag{B.4}$$

where $v_0 = \mu_2^2/\sqrt{2\lambda(\mu_1^2 + \mu_2^2)} = v(0)$. Note that, if $v(x)$ is a solution, then $-v(x)$ is also a solution because of the $\mathbb{Z}_2$ symmetry. We can work with one of them without loss of generality.

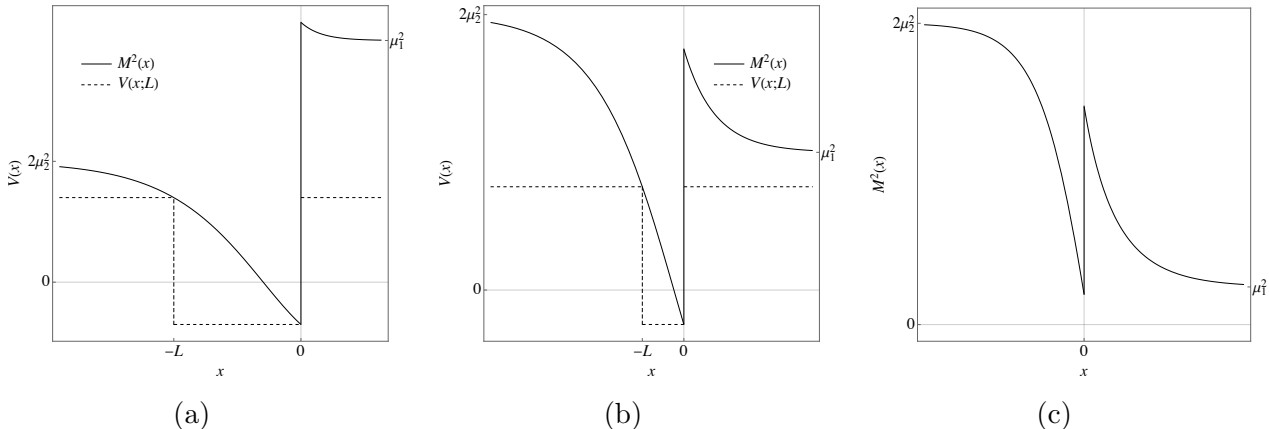

(a)                          (b)                          (c)

Figure 8: The potential $M^2(x)$ (solid line) and their corresponding lower bounding square-well potentials $V(x; L)$ (dotted line) when: (a) $\mu_2^2 < \mu_1^2/2$, (b) $\mu_1^2/2 < \mu_2^2 < 2\mu_1^2$, and (c) $\mu_2^2 > 2\mu_1^2$.

Consider the linearized fluctuation around this vacuum $\Phi = v + \varepsilon\varphi$, which satisfies

$$-\partial^2\varphi + M^2(x)\varphi = 0 \ , \tag{B.5}$$

where $M^2(x) = \mu^2(x) + 3\lambda v^2(x)$. Since the linearized equation of motion is translational invariant along the transverse coordinates $(\vec{y}, t)$, we can assume the following ansatz

$$\varphi = e^{iEt - i\vec{p}\cdot\vec{y}}\psi(x) \ . \tag{B.6}$$

The wave function $\psi(x)$ is the eigenfunction of a one-dimensional Schrdinger equation

$$-\partial_x^2\psi(x) + M^2(x)\psi(x) = m^2\psi(x) \ , \tag{B.7}$$

whose eigenvalue $m^2 = E^2 - |\vec{p}|^2$ is the effective 2+1 dimensional mass of the excitations. The eigenvalue $m^2$ is always non-negative if $v(x)$ is the true vacuum with minimal energy. To show that it is indeed positive, i.e., there are no zero modes, consider the following three cases (see figure 8):

- $\mu_2^2 < \mu_1^2/2$: In this case, $0 < M^2(-\infty) < M^2(+\infty)$, and we have a region where $M^2(x) < 0$. The negative $M^2(x)$ region lies in $-L_0 < x < 0$ where $L_0 > 0$ depends on the parameters $(\mu_1, \mu_2$ and $\lambda)$ of the theory. In fact, the minimum is obtained at $x = 0$, where $M^2(0^-) = -\mu_2^2 + 3\lambda v_0^2 < 0$. For some $L > L_0$, consider a square-well potential of the form

$$V(x; L) = \begin{cases} M^2(-L) \ , & x < -L \text{ or } x > 0 \\ M^2(0^-) \ , & -L < x < 0 \end{cases} \tag{B.8}$$

This potential is chosen such that $M^2(x) \geq V(x; L)$ for all $x$ and $L$, i.e., the square-well potential *lower bounds* $M^2(x)$. Hence, if we can show that there exists an $L$ such that the Schrödinger equation (B.7) with potential $V(x; L)$ has only positive eigenvalues, then the Schrödinger equation with potential $M^2(x)$ also has only positive eigenvalues. First, it is clear that $L > L_0$ for the square-well potential to have any positive eigenvalues with bounded solutions at all. As is familiar from quantum mechanics, the eigenvalues of the square-well problem are given by a transcendental equation,

$$\lambda = -M_1^2 + \frac{4\eta^2}{L^2}, \quad \text{where } \eta \sec \eta = \frac{M_0(L)L}{2}, \tag{B.9}$$

and $M_0^2(L) = M^2(-L) + M_1^2 > 0$ is the *depth* of the potential $V(x; L)$, and $M_1^2 = -M^2(0^-) > 0$. The lowest eigenvalue corresponds to $0 < \eta < \frac{\pi}{2}$. Requiring $\lambda > 0$ then gives us the constraint that

$$\eta > \frac{M_1 L}{2} \implies \frac{M_0(L)L}{2} = \eta \sec \eta > \frac{M_1 L}{2} \sec \frac{M_1 L}{2} \implies \frac{M^2(-L)}{M_1^2} > \tan^2 \frac{M_1 L}{2}, \tag{B.10}$$

because $\sec x$ is a monotonically increasing function in $0 < x < \frac{\pi}{2}$. Since $\eta < \frac{\pi}{2}$, we only need to look for $L < \pi/M_1$. Therefore, the possible range of $L$ is $L_0 < L < \pi/M_1$. Defining $\ell = \frac{M_1 L}{2}$ and $\ell_0 = \frac{M_1 L_0}{2}$, we can write the range as $\ell_0 < \ell < \frac{\pi}{2}$, and the constraint as

$$\frac{M^2(-2\ell/M_1)}{M_1^2} > \tan^2 \ell. \tag{B.11}$$

For $\mu_2^2 = \mu_1^2/2$, there is indeed an $\ell$ that satisfies the above constraint (for example, choosing $L$ such that $\mu_{4d}^2(-L) = \frac{3}{4}\mu_1^2$ satisfies the above condition). As $\mu_2/\mu_1 \to 0$, the LHS approaches $3 \tanh^2 \sqrt{2}\ell - 1$, and it can be verified numerically that any $\ell$ between $\ell_1 = 0.647549$ and $\ell_2 = 0.730573$ satisfies (B.11). In fact, as the above two limits suggest, for the entire range of $\mu_2$ considered here, an $\ell$ that satisfies (B.11) can always be found. Figure 9 shows these limiting cases and some cases in between. Therefore, any bounded solution of (B.7) has $m^2 > 0$.

- $\mu_1^2/2 < \mu_2^2 < 2\mu_1^2$: In this case, there is still a region where $M^2(x) < 0$ but now $0 < M^2(+\infty) < M^2(-\infty)$. Once again, we can use a *lower bounding* square-well potential

$$V(x; L) = \begin{cases} \frac{3}{4}\mu_1^2, & x < -L \text{ or } x > 0 \\ M^2(0^-), & -L < x < 0 \end{cases} \tag{B.12}$$

where $L$ is chosen such that $M^2(-L) = \frac{3}{4}\mu_1^2$ (here, $\frac{3}{4}$ isn't special, it just serves the purpose). It can be shown that for any $\mu_2$ in the given range, this potential has only positive eigenvalues. Hence, even in this case, any bounded solution of (B.7) has

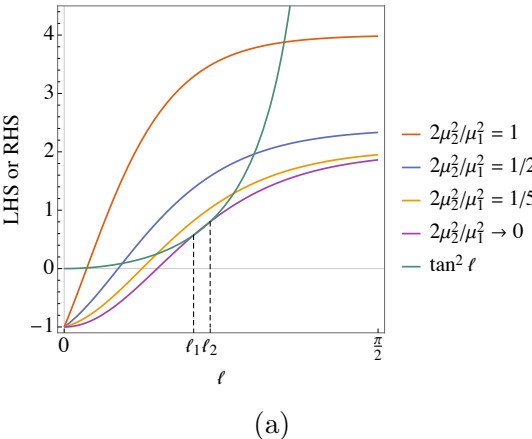 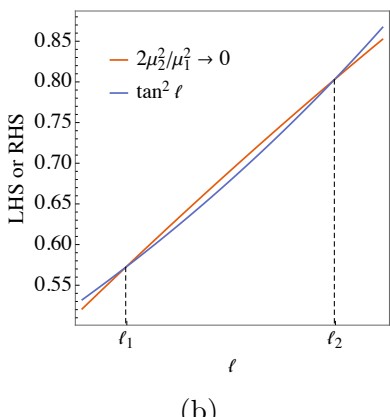

(a)           (b)

Figure 9: (a) Four cases of the LHS of equation (B.11) are plotted here. All of them have a range of $\ell$ where they are above the RHS ($\tan^2 \ell$) curve. (b) Zoomed up version of $\mu_2/\mu_1 \to 0$, in which case the LHS of (B.11) corresponds to $3 \tanh^2 \sqrt{2}\ell - 1$.

$m^2 > 0$.

- $\mu_2^2 > 2\mu_1^2$: In this case, $M^2(x) > 0$ everywhere, and hence, any bounded solution of (B.7) has $m^2 > 0$.

We conclude that the effective 2+1 dimensional mass $m^2$ of the $\eta'$ excitations are always positive when $\mu^2(x)$ is discontinuous. Although the proof is restricted to discontinuous $\mu^2(x)$, we expect that deforming $\mu^2(x)$ to a generic profile doesn't change the above conclusion drastically.

It is interesting to consider the limit $\mu_{1,2}, \lambda \to \infty$ such that $\mu_{1,2}/\sqrt{\lambda}$ is finite. In this limit, the vacuum solution (B.4) behaves like a step function with $v(x) = \mu_2/\sqrt{\lambda}$ for $x < 0$, and $v(x) = 0$ for $x > 0$. In this case, the system can be thought of as being in the massive phase on one side, and the condensed phase on the other side.

## B.2    String

Let us now go back to our original problem with both $\kappa(y)$ and $\mu^2(x)$ being discontinuous step functions. We define $\mu^2(x)$ as before, and $\kappa(y)$ as an *odd* step function with $\kappa(y) = \kappa_0$ for $y > 0$. The vacuum $v(x, y)$ satisfies the equation of motion

$$- (\partial_x^2 + \partial_y^2)v(x,y) + \kappa(y) + \mu^2(x)v(x,y) + \lambda v(x,y)^3 = 0 . \tag{B.13}$$

Due to the $\mathbb{Z}_2$ symmetry, $y \to -y$ and $\Phi \to -\Phi$, of the Lagrangian (4.12), we expect the vacuum to be an odd function of $y$. In particular, this means $v(x, 0) = 0$ for all $x$. At large $|x|$, we expect the vacuum to be independent of $x$, so the equation (B.13) reduces to

a one-dimensional problem in $y$. An exact solution for both $v(+\infty, y)$ and $v(-\infty, y)$ can be found with suitable boundary conditions. Since $\mu^2(-\infty) < 0 < \mu^2(+\infty)$, the asymptotic solutions satisfy $v^2(-\infty, y) \geq v^2(+\infty, y)$ for all $y$. We expect the vacuum to interpolate between these two asymptotic solutions monotonically, so $v^2(x, y) \geq v^2(+\infty, y)$ for all $x$ and $y$.

Assuming the ansatz $\varphi = e^{iEt-ipz}\psi(x, y)$ for the linearized fluctuation around the vacuum $\Phi = v + \varepsilon\varphi$, the *wave function* $\psi(x, y)$ is the eigenfunction of a two-dimension Schrödinger equation

$$-(\partial_x^2 + \partial_y^2)\psi(x, y) + M^2(x, y)\psi(x, y) = m^2\psi(x, y) ,\qquad\text{(B.14)}$$

where $M^2(x, y) = \mu^2(x) + 3\lambda v^2(x, y)$, and the eigenvalue $m^2 = E^2 - p^2$ is the effective 1+1 dimensional mass of the excitations. Since $v(x, 0)$ is always zero, the potential $M^2(x, 0)$ is negative for $x < 0$. In fact, there is always a region around the negative $x$ axis where $M^2(x, y) < 0$. Hence, there is always a possibility of bounded solutions with zero eigenvalues $m^2$. We shall show that this is not the case.

Consider the following lower bound on the potential,

$$M^2(x, y) = \mu^2(x) + 3\lambda v^2(x, y) \geq -\mu_2^2 + 3\lambda v^2(+\infty, y) .\qquad\text{(B.15)}$$

Although this is a crude bound, it is enough for our purposes for a wide range of $\mu_1, \mu_2, \lambda$ and $\kappa_0$. The RHS in the above inequality can be further *lower bounded* by the following square-well potential,

$$V(y; L) = \begin{cases} -\mu_2^2 , & |y| < L \\ -\mu_2^2 + 3\lambda v^2(\infty, L) , & |y| > L \end{cases}\qquad\text{(B.16)}$$

Let $u$ be the real root of the cubic polynomial $u^3 + u + (\sqrt{\lambda}\kappa_0/\mu_1^3) = 0$ (there is only one real root). It can be shown that, whenever $u^2 \geq 2\mu_2^2/\mu_1^2$, there exists a value of $L$ such that the eigenvalues of this square-well problem are positive (here, the factor of 2 in the inequality can be made smaller by more careful analysis). Therefore, as long as this condition is met, the above crude bound allows us to conclude that the eigenvalues $m^2$ of the Schrödinger equation (B.14) are always positive.

Once again, the above proof shows that the effective 1+1 dimensional mass $m^2$ of the $\eta'$ excitations are always positive in the case of a discontinuous $\kappa(y)$ and $\mu^2(x)$, *and* with a certain restriction on the parameters. We expect that, with better lower bounds on the potential $M^2(x, y)$, we can relax this condition but we do not attempt this exercise here. Moreover, deforming $\kappa(y)$ and $\mu^2(x)$ to a generic profile shouldn't change the above conclusion drastically.

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
