# Peer review of "Interface Junctions in QCD${}_4$"

_SciPost Physics_

## Round 2 · Referee Report · Anonymous (Referee 1) · 2020-9-7

Report

The present manuscript deals with interfaces of CP symmetry breaking in 4 dimensional gauge theories with $N_f$ flavors, as parameters of the theory vary in space. In taking the common mass of the quarks to be essentially proportional to the complex parametrization of a plane section of physical space, the theory displays $N_f$ 2+1 dimensional interfaces that converge radially to a 1+1 dimensional junction around the origin of the plane. The theories on each interface away from the junction are known from previous work (where only the $\theta$ angle was varying). The main result of this manuscript is to determine the theory on the 1+1 dimensional junction, based on symmetry breaking arguments and anomaly inflow. Accordingly, the paper includes a careful analysis of the anomalies of the theories involved, and how they localize on their boundaries. This analysis is done in all the regimes involved, since by construction the mass scale of the flavors changes with the radius and hence scans all the possible hierarchies with respect to the strong coupling scales of the various theories (both bulk and interface).

I found the paper well structured, complete in its details, and stimulating in the physics discussed. Therefore I think that it certainly deserves to be published. However I would like the authors to address a few points that I list below, in order to improve the presentation and remove a few obscure issues.

1) Just the sentence before eq. (2.11), I believe the correct version is that the WZ term does not depend on the extension. 2) The notation in eqs. (2.12) and (2.13) is perhaps not clear for the non-experts: where are sums implied? 3) At the very end of section 5, there is no discussion of the $\epsilon\sim 1$ case for $N_f=2$. If there is no difference w.r.t. the $\epsilon \ll 1$ case, it should be at least mentioned. 4) Perhaps related to the previous point, at the beginning of section 6, there is a quick sentence saying that for a general winding mass profile, in the $N_f=2$ case the two interfaces are connected by a 1+1 dimensional junction at the origin. This is surprising, because at least for $\epsilon\ll 1$, I would expect something very similar to the `straight' situation of the previous section. Some clarification is needed in my opinion. 5) For the appendices, my humble suggestion is the following. Perhaps it could help the reader to invert their order, since in Appendix B the focus is on the scalar profile that is then instrumental in the discussion of Appendix A. Alternatively, the authors could refer to section B.1 in the first paragraph of section A.1, when $v(x)$ is introduced. 6) Finally, the manuscript needs to be read through carefully, since I have spotted some trivial typos (one every couple of pages, on average) that can be easily eliminated.

  • validity: -
  • significance: -
  • originality: -
  • clarity: -
  • formatting: -
  • grammar: -

Author:  Ho Tat Lam  on 2020-09-09  [id 954]

(in reply to Report 1 on 2020-09-07)

Thanks for the comments and suggestions! See the reply below.

  1. We will fix it.
  2. We are summing over indices a,b,c,d running from 1 to $N_f$. We will add a summation symbol to clarify it.
  3. Yes, the discussion for $\epsilon\gg1$ is still valid at $N_f=2$. We will add a sentence to clarify it.
  4. Yes, the discussion for the "straight" case still apply. A minor difference is that the two interfaces now have an angle. We will rewrite the sentence.
  5. We prefer to keep the original ordering as how they appear in the main text. We will add a sentence in appendix A referring to appendix B as you suggested.
  6. We will go through the draft and try to eliminate as many typos as possible.

Author:  Ho Tat Lam  on 2020-09-09  [id 953]

(in reply to Report 1 on 2020-09-07)

Thanks for the comments and suggestions! See the reply below.

  1. We will fix it.
  2. We are summing over indices a,b,c,d running from 1 to $N_f$. We will add a summation symbol to clarify it.
  3. Yes, the discussion for $\epsilon\gg1$ is still valid at N=2. We will add a sentence to clarify it.
  4. Yes, the discussion for the "straight" case still apply. A minor difference is that the two interfaces now have an angle. We will rewrite the sentence.
  5. We prefer to keep the original ordering as how they appear in the main text. We will add a sentence in appendix A referring to appendix B as you suggested.
  6. We will go through the draft and try to eliminate as many typos as possible.

---

## Round 2 · Referee Report · Anonymous (Referee 2) · 2020-9-15

Strengths

Open a new pathway in an existing research direction

Report

The paper studies four-dimensional QCD in the regime where the number of flavors $N_F$ is smaller than $N_{CFT}$, so that the infrared behavior is described by a chiral Lagrangian.

The papers focuses on three-dimensional interfaces and two-dimensional junctions of interfaces. The authors propose a description of such junctions. This is a new and important non-perturbative result about four-dimensional QCD. So the paper deserves to be published on SciPost Physics.

I think that their way of attacking this problem can be generalized to other examples of strongly coupled QFT's.

The authors provide the relevant background needed for their arguments, which are also explained very clearly (except for quite a few a typos that should be fixed).

---

## Editorial Decision

resubmitted